# Reward-Shaping Control Variates For Off-Policy Evaluation Under Sparse Rewards

Ritam Majumdar [* 1]   Finale Doshi-Velez [2]   Sonali Parbhoo [* 1]

## Abstract

Off-policy evaluation (OPE) is essential for deploying reinforcement learning in safety-critical settings, yet existing estimators such as importance sampling and doubly robust (DR) often exhibit prohibitively high variance when rewards are sparse. In this work, we introduce Reward-Shaping Control Variates, a new family of unbiased estimators that leverage potential-based reward shaping to construct additional zero-mean control variates. We prove that shaped estimators always yields valid variance reduction, and that combining shaping-based and Q-based control variates strictly expands the variance-reduction subspace beyond DR and its minimax variant MRDR. Empirically, we provide a systematic regime map across synthetic chains, a cancer simulator, 5 single-stock and 1 multi-stock DOW-30 trading environments and an ICU-sepsis benchmark showing that shaping-based OPE consistently outperforms DR in sparse-reward settings, while a hybrid estimator achieves state-of-the-art performance across sparse, noisy, and misspecified environments. Our results highlight reward shaping as a powerful and interpretable tool for robust OPE, offering both theoretical guarantees and practical improvements in domains where standard estimators fail.

## 1. Introduction

Off-policy evaluation (OPE) is a central problem in reinforcement learning (RL): given data collected by a *behavior policy*, the goal is to estimate the performance of a different *evaluation policy* without deploying it in the environment. Reliable OPE is crucial in high-stakes applications such as healthcare, education, and recommendation systems, where

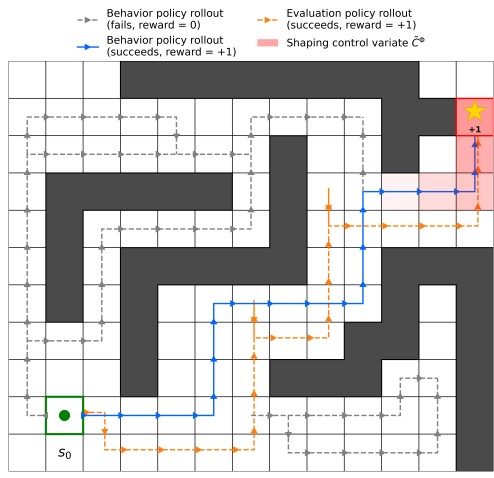

*Figure 1.* Sparse-reward maze example. Most behaviour rollouts (grey) fail and receive 0 reward; only 1% (blue) succeed with +1. The evaluation policy succeeds 10%, but off-policy estimates (orange) require thousands of samples to stabilize due to the scarcity of reward in behavior data. Shaped estimators leverage additional signal (red), capturing progress to success, as a control variate for variance reduction in OPE.

deploying untested policies can have significant costs. Standard OPE methods fall broadly into two families: importance sampling (IS) (Precup et al., 2000; Jiang & Li, 2016), which reweights trajectories to mimic the evaluation policy, and doubly robust (DR) methods (Dudik et al., 2011; Thomas & Brunskill, 2016), which combine reweighting with regression-based reward prediction. These estimators are well studied and can perform effectively when rewards are dense and trajectories contain frequent signals. However, many real-world settings are characterized by *sparse rewards*, where meaningful feedback occurs only rarely or at the end of a long horizon. In such cases, existing estimators become unreliable.

To see why, consider a maze with 100 decision steps where the agent receives +1 only upon reaching the goal. Suppose the behavior policy succeeds 1% of the time while the evaluation policy succeeds 10%. Nearly all logged trajectories contain *zero reward*; only rare successes provide information about the policy's value. This illustrates the core challenge in sparse-reward OPE: even if the evaluation

[*]Equal contribution  [1]Imperial College London [2]Harvard University. Correspondence to: Sonali Parbhoo <sparbhoo@ic.ac.uk>.

*Proceedings of the 43rd International Conference on Machine Learning*, Seoul, South Korea. PMLR 306, 2026. Copyright 2026 by the author(s).

policy succeeds frequently, the logged data may contain almost no trajectories revealing why success happens, making it difficult for any estimator to propagate reward signal backward through long horizons.

IS collapses under this regime: non-zero rewards arise only on rare successful trajectories, so the estimate is dominated by a tiny fraction of samples with potentially enormous importance weights, causing variance to grow prohibitively large. DR methods fail for a different reason. DR introduces a learned Q-function to reduce variance by filling in returns where rewards are not observed. But under sparsity, most temporal-difference targets are zero, providing almost no supervision for learning long-horizon values. *The model must extrapolate the value of reaching the goal from very few successful rollouts* and any systematic error in this extrapolation becomes bias in the final estimate, persisting even when the IS component is unbiased.

Marginalized estimators such as AvgDICE (Che et al., 2025), DualDICE (Nachum et al., 2019) and GenDICE (Zhang et al., 2020) reduce variance by estimating state(-action) distribution ratios rather than trajectory weights. While effective in moderately sparse domains, they remain brittle under extreme sparsity: their optimization is anchored to reward support, so when nearly all observed rewards are zero, many density ratios become equally consistent with the data, making the objective ill-conditioned. This yields estimates that appear stable but converge to incorrect values leading to low variance but high bias. Moreover, when the evaluation policy succeeds in regions the behavior policy rarely visits, marginalized estimators inherit the same extrapolation gap as DR methods (Uehara et al., 2020).

In this paper, we address this challenge by introducing a new class of *shaped estimators* that reduce variance without introducing bias. Our key observation is that while rewards are sparse, *state visitations are not*: every trajectory, successful or not, visits a dense sequence of states. If we can distinguish states that "on the path to success" from those that "drift towards failure", we can extract signal from all trajectories, not just the rare successful ones. We operationalize this through *potential-based reward shaping*. Specifically, we learn potential functions $\Phi(s_t)$ that assign a scalar "progress score" to each state $s_t$: high for states appearing on successful trajectories, low for those on failed ones. Crucially, by construction these potential functions *do not predict rewards or modify the evaluation target*. Instead, we use the importance-weighted shaping differences $W_t(\gamma\Phi(s_{t+1}) - \Phi(s_t))$ as a *control variate*: a zero-mean correction term correlated with the noisy IS estimate.

By construction, this shaping term has zero expectation under the behavior-policy data distribution due to the telescoping structure of potential-based shaping. As a result, adding it preserves unbiasedness *independently of how $\Phi$*

*is learned.* Moreover, because the control variate is correlated with the IS estimator's noise, subtracting its optimal amount is guaranteed to reduce variance. This mechanism fundamentally differs from existing approaches. Unlike doubly robust (DR) estimators, which must accurately predict sparse rewards, our potential $\Phi$ is learned from dense state visitation patterns. Unlike marginalized or ratio-learning estimators, which require reward signal to identify which distribution ratios matter, our method injects this structure directly via shaping. The result is an estimator that remains unbiased by construction while extracting variance-reducing signal from every trajectory, including the overwhelming majority that never reach the goal.

**Contributions.** Our contributions are as follows: 1. We introduce a theoretical framework showing how potential-based reward shaping can be leveraged as a control variate in OPE, proving that the resulting estimators remain unbiased while achieving strict variance reduction over standard IS, DR, and marginalized estimators. 2. We analyze sparse and noisy reward regimes, identifying conditions under which shaping control variates provide significant improvements. 3. We demonstrate across tabular MDPs, a cancer simulator (Ribba et al., 2012), single and multi-stock DOW-30 trading environments, and an ICU-Sepsis benchmark (Choudhary et al., 2024) that shaped estimators consistently outperform existing baselines. 4. We provide practical guidance on constructing shaping functions, highlighting trade-offs between variance reduction and robustness. Together, these contributions establish reward-shaping control variates as a principled and practical solution for robust OPE where standard estimators fail.

## 2. Related Work

**IS, DR and Marginalized Estimators.** A well-established body of work in OPE includes IS (Precup et al., 2000; Jiang & Li, 2016) and DR methods (Dudik et al., 2011; Thomas & Brunskill, 2016). IS reweights trajectories but suffers exponential variance under sparsity, as rare successes dominate importance weights. DR learns a Q-function as a control variate, but Q must be learned from reward signal under sparsity, this requires extrapolation that introduces bias. Marginalized estimators such as AvgDICE (Che et al., 2025), GenDICE (Zhang et al., 2020), and VPM (Uehara et al., 2020) estimate state-visitation ratios, avoiding exploding importance products, but still rely on reward signal to identify which states matter; under extreme sparsity, density-ratio optimization becomes ill-conditioned, yielding stable but biased estimates. Our approach learns a potential $\Phi(s)$ but uses it fundamentally differently. Rather than predicting rewards, $\Phi$ induces shaping differences $\gamma\Phi(s') - \Phi(s)$ that telescope to a boundary term regardless of what $\Phi$ is learned. This telescoping property (Ng et al., 1999) guarantees zero-

mean control variates under any policy even when $\Phi$ is optimized for variance reduction, yielding an estimator that remains unbiased by construction while extracting signal from dense state visitations rather than sparse rewards.

**Model-Based and Representation-Learning Approaches.** Another line of OPE research leverages model-based roll-outs (Le et al., 2019) or representation learning (e.g., invariant or confounder-aware embeddings) to mitigate distributional shifts (Hanna et al., 2017; Uehara et al., 2022).These however, require accurate dynamics or strong sufficiency assumptions. Our method sidesteps this by reshaping rewards rather than modeling dynamics. Majumdar et al. (2025) incorporate human-interpretable "concepts" to reduce variance while remaining unbiased, assuming concept supervision or the ability to discover reflective abstractions. While also targeting variance reduction, their method depends on concept supervision, whereas ours directly reshapes rewards, enhancing evaluation fidelity even in standard RL settings with known reward functions.

**Reward Shaping in Reinforcement Learning.** Potential-based reward shaping accelerates learning without changing optimal policies (Ng et al., 1999), and has been used to improve sample efficiency in online RL (Müller & Kudenko, 2025; Devlin & Kudenko, 2012; Grześ, 2017). Prior OPE work considered fixed shaping-based control variates (Parbhoo et al., 2020), but only in limited settings. Unlike Parbhoo et al. (2020), we explicitly learn zero-mean control variates for variance reduction, prove unbiasedness and variance reduction for varying strengths of the control variates and also provide an algorithm that jointly optimizes the potential function and its strength to guarantee variance reduction. This generalization strictly subsumes Parbhoo et al. (2020) and achieves state-of-the-art performance in sparse, noisy, and misspecified environments.

## 3. Preliminaries and Notation

**Markov Decision Processes.** A Markov decision process (MDP) is a tuple $M = (\mathcal{S}, \mathcal{A}, P, \gamma, R)$, where $\mathcal{S}$ is the state space, $\mathcal{A}$ the action space, $P(s, a, s')$ the transition distribution from $s$ to $s'$ under action $a$, $R(s, a)$ the reward for $(s, a)$, and discount factor $\gamma$. A policy $\pi : \mathcal{S} \times \mathcal{A} \to [0, 1]$ maps states to action probabilities, with $\pi(a|s)$ the probability of taking $a$ in $s$. A trajectory $\tau = (s_0, a_0, r_0, \dots, s_T)$ generated by $\pi$ has return $R_{0:T-1}(\tau) = \sum_{t=0}^{T-1} \gamma^t r_t$, where $s_{t+1} \sim P(\cdot|s_t, a_t)$ and $a_t \sim \pi(\cdot|s_t)$. The performance of $\pi$ is $V^\pi = \mathbb{E}_{P_\tau^\pi}[R_{0:T-1}(\tau)]$. The value and action-value functions, $V^\pi(s)$ and $Q^\pi(s, a)$, are the expected returns starting from $s$, or from $(s, a)$ followed by $\pi$, respectively.

**Off-Policy Evaluation.** Given a dataset $\mathcal{D} = \{\tau^{(i)}\}_{i=1}^n$ of trajectories generated by a *behavior policy* $\pi_b$, we wish to estimate the value of a different *evaluation policy* $\pi_e$. A valid estimator $\hat{V}^{\pi_e}$ should minimize the mean squared error (MSE):

$$\text{MSE}(V^{\pi_e}, \hat{V}^{\pi_e}) = \underbrace{\left(\mathbb{E}_{P_\tau^{\pi_b}}[\hat{V}^{\pi_e}] - V^{\pi_e}\right)^2}_{\text{Bias}^2} + \underbrace{\text{Var}(\hat{V}^{\pi_e})}_{\text{Variance}}.$$

where $P_\tau^{\pi_b}$ denotes the distribution of trajectory $\tau$ under behaviour policy $\pi_b$. This decomposition highlights that reducing variance while maintaining unbiasedness is central to designing effective OPE estimators. We adopt the following standard assumptions:

**Assumption 1 (Absolute Continuity).** For all $(s, a) \in \mathcal{S} \times \mathcal{A}$, if $\pi_b(a \mid s) = 0$, then $\pi_e(a \mid s) = 0$.

**Assumption 2 (Single Behavior Policy).** All trajectories in $\mathcal{D}$ are sampled independently under the same behavior policy $\pi_b$.

**Per-step IS estimator (Precup et al., 2000).** Let $Y_t := \gamma^t W_t r_t$ denote the per-step IS reward where $W_t = \prod_{k=0}^t \frac{\pi_e(a_k|s_k)}{\pi_b(a_k|s_k)}$ is the cumulative importance weight up to time $t$. For a single trajectory $\tau \sim \pi_b$, the per-step IS return is given by $Y := \sum_{t=0}^{T-1} Y_t$. The per-step IS estimator of the return is given by,

$$\hat{V}_{\text{PDIS}}^{\pi_e} = \frac{1}{N} \sum_{n=1}^N Y^{(n)} = \frac{1}{N} \sum_{n=1}^N \sum_{t=0}^{T-1} \gamma^t W_t r_t \quad (1)$$

**Potential-Based Reward Shaping (Ng et al., 1999).** Reward shaping is a technique that is used to modify the original reward function using a reward-shaping function $F : \mathcal{S} \times \mathcal{A} \times \mathcal{S} \to \mathbb{R}$ to typically make RL methods converge faster with more instructive feedback. The original MDP $M = (\mathcal{S}, \mathcal{A}, P, \gamma, R)$ is transformed into a *shaped-MDP $M' = \mathcal{S}, \mathcal{A}, P, \gamma, R' = R + F$*). Reward shaping modifies the original reward signal using an auxiliary shaping function $F : \mathcal{S} \times \mathcal{A} \times \mathcal{S} \to \mathbb{R}$, yielding a new reward function

$$R'(s, a, s') = R(s, a) + F(s, a, s').$$

While arbitrary shaping may alter the optimal policy, *potential-based reward shaping (PBRS)* (Ng et al., 1999) preserves policy invariance.

**Definition 3.1 (Potential-Based Reward Shaping).** A shaping function $F$ is potential-based if there exists a potential $\Phi : \mathcal{S} \to \mathbb{R}$ such that

$$F^\Phi(s, a, s') = \gamma \Phi(s') - \Phi(s), \quad \forall (s, a, s') \in \mathcal{S} \times \mathcal{A} \times \mathcal{S}.$$

**Theorem 3.2 (Policy Invariance under PBRS).** *If shaping is potential-based, then for all policies $\pi$, the optimal policy under the shaped MDP coincides with the optimal policy under the original MDP. That is, PBRS modifies value functions by a state-dependent constant shift but leaves the optimal action distribution unchanged.*

This invariance property makes PBRS especially appealing for OPE: it allows us to introduce additional signal into sparse-reward environments while guaranteeing that the evaluation policy's value is estimated consistently with the original reward structure. As the learnt shaping potentials depend only on states, there is no modification to the policy, thus maintaining the fidelity of the task.

## 4. Reward-Shaping Control Variates

In this section we introduce a new class of *reward-shaping control variates* for variance reduction in off-policy evaluation. The key idea is to construct an additional random variable $C^\Phi$ with zero mean under the behavior-policy distribution. Such a control variate can be added to any unbiased OPE estimator without altering its expectation, but with the potential to substantially reduce variance. We first formalize the construction of $C^\Phi$ and then show how it can be integrated into the per-decision importance sampling (PDIS) estimator. While we present results for PDIS, the same construction can be incorporated into other OPE estimators (e.g., weighted IS, DR, MRDR) with analogous guarantees. See Appendix B for details.

**Terminal boundary condition.**    Let $\Phi : \mathcal{S} \to \mathbb{R}$ be any measurable potential function such that $\Phi(s_T) = 0$ for all terminal states $s_T$. As we know the reward outcome at the final timestep, we do not have to shape the reward at states which occur at the final timestep $T$. Importantly, trajectories may terminate in different terminal states depending on outcomes (e.g., survival vs. death in ICU, different final portfolios in trading); the condition does not require terminal states to be identical, but simply fixes the potential value to zero on the terminal set. This anchoring is standard in potential-based shaping and ensures that the shaping differences telescope i.e. $\sum_{t=0}^{T-1} \gamma^t(\gamma\Phi(s_{t+1}) - \Phi(s_t)) = \gamma^T \Phi(s_t) - \Phi(s_0) = -\Phi(s_0)$.

**Defining Potential-Based Shaping Features.**    Define per-step shaping potential differences $F_t^\Phi = \gamma\Phi(s_{t+1}) - \Phi(s_t) \ \forall t = 0, \ldots T - 1$. For each trajectory $\tau \sim \pi_b$, we define a *shaping feature vector*,

$$C^\Phi(\tau) := (C_{-1}^\Phi, C_0^\Phi, \ldots, C_{T-1}^\Phi)^\top \in \mathbb{R}^{T+1}, \quad (2)$$

with components $\forall t = 0, \ldots, T - 1$

$$C_t^\Phi := \gamma^t W_t F_t^\Phi = \gamma^t W_t(\gamma\Phi(s_{t+1}) - \Phi(s_t)) \quad (3)$$

$$C_{-1}^\Phi := \Phi(s_0). \quad (4)$$

### 4.1. Potential-Based Reward Shaping Estimators for OPE

We will show how to use the potential-based shaping features from Eq. 3 to construct control variates for off-policy evaluation. Specifically, these shaping features can be used to construct zero-mean control variates under the data-generating distribution which preserve unbiasedness, while enabling substantial variance reduction when they correlate with the IS-weighted return.

**Definition 4.1 (Shaped PDIS estimator).** Given $N$ i.i.d. trajectories, the shaped PDIS estimator is defined as

$$\widehat{V}_{\text{Shaped-PDIS}}^{\pi_e}(\lambda) := \frac{1}{N}\sum_{n=1}^{N} Y^{(n)} + \lambda^\top \widetilde{C}^{\Phi,(n)},$$

where $\tilde{C}$ denotes a centered version of the shaping feature vector and $\lambda \in \mathbb{R}^{T+1}$ is a coefficient vector controlling the weight of the centered shaping feature vector. Specifically, we center each feature dimension under the behavior policy distribution, $\widetilde{C}^\Phi := C^\Phi - \mathbb{E}_{\pi_b}[C^\Phi]$.

**Lemma 4.2 (A Reward-Shaping Control Variate).** *Under Assumption 1 and the boundary condition $\Phi(s_T) = 0$, the centered shaping feature vector $\tilde{C}^\Phi$ is a valid zero-mean control variate for the PDIS return.*

*Proof:* By construction, centering $C^\Phi$ subtracts the mean of $\pi_b$ from each component. Thus,

$$\mathbb{E}_{\pi_b}[\widetilde{C}^\Phi] = \mathbb{E}_{\pi_b}[C^\Phi] - \mathbb{E}_{\pi_b}[C^\Phi] = 0. \quad \square$$

### 4.2. Theoretical Guarantees of Reward Shaping Control Variates

In what follows, we provide theoretical guarantees for Shaped-PDIS. Note that analogous theoretical guarantees can be derived for other shaped estimators. Details of these and their proofs can be found in Appendix C.1.

**Theorem 4.3 (Unbiasedness of Shaped PDIS).** *For any $\lambda \in \mathbb{R}^{T+1}$, the Shaped PDIS estimator is unbiased. That is, $\mathbb{E}_{\pi_b}\left[\widehat{V}_{\text{Shaped-PDIS}}^{\pi_e}(\lambda)\right] = V^{\pi_e}$.*

*Proof.* By linearity of expectation,

$$\mathbb{E}_{\pi_b}\left[\widehat{V}_{\text{Shaped-PDIS}}^{\pi_e}(\lambda)\right] = \mathbb{E}_{\pi_b}\left[\widehat{V}_{\text{PDIS}}^{\pi_e}\right] + \mathbb{E}_{\pi_b}\left[\lambda^\top \tilde{C}^\Phi\right]$$
$$= V^{\pi_e}$$

From IS theory, $\mathbb{E}_{\pi_b}[\widehat{V}_{\text{PDIS}}^{\pi_e}] = V^{\pi_e}$ and by Lemma 4.2, $\mathbb{E}_{\pi_b}[C^\Phi] = 0$. Thus, the estimator is unbiased for any choice of $\lambda$. $\square$

Next, we characterize the variance of $\widehat{V}_{\text{Shaped-PDIS}}^{\pi_e}(\lambda)$ and discuss the optimal choice of $\lambda$ that minimizes the variance, $\lambda^\star$. Recall that the shaped estimator has the form $Y + \lambda^\top \widetilde{C}^\Phi$. Since the trajectories $\tau$ are i.i.d., $\text{Var}\left[\widehat{V}_{\text{Shaped-PDIS}}(\lambda)\right] = \frac{1}{N}\text{Var}(Y + \lambda^\top \widetilde{C}^\Phi)$. Let

$$\boldsymbol{\Sigma}_{CC} := \text{Cov}(\widetilde{C}^\Phi, \widetilde{C}^\Phi) \in \mathbb{R}^{(T+1)\times(T+1)}, \quad (5)$$

$$\boldsymbol{\Sigma}_{CY} := \text{Cov}(\widetilde{C}^\Phi, Y) \in \mathbb{R}^{T+1}. \quad (6)$$

**Theorem 4.4** (**Variance-optimal** $\lambda^*$). *Assume* $\Sigma_{CC}$ *is positive definite. The variance-minimizing coefficients are*

$$\lambda^\star = -\Sigma_{CC}^{-1}\Sigma_{CY}, \qquad (7)$$

*and the corresponding minimized variance is*

$$\mathrm{Var}\Big(Y + \lambda^{\star\top}\widetilde{C}^\Phi\Big) = \mathrm{Var}(Y) - \Sigma_{CY}^\top\Sigma_{CC}^{-1}\Sigma_{CY}. \quad (8)$$

*Proof.* Expand

$$\mathrm{Var}\Big(Y + \lambda^\top\widetilde{C}^\Phi\Big) = \mathrm{Var}(Y) + 2\lambda^\top\Sigma_{CY} + \lambda^\top\Sigma_{CC}\lambda.$$

This is a convex quadratic in $\lambda$. Setting its gradient $2\Sigma_{CY} + 2\Sigma_{CC}\lambda$ to zero yields $\lambda^\star = -\Sigma_{CC}^{-1}\Sigma_{CY}$. Substituting $\lambda^\star$ back gives $\mathrm{Var}(Y + \lambda^{\star\top}\widetilde{C}^\Phi) = \mathrm{Var}(Y) - \Sigma_{CY}^\top\Sigma_{CC}^{-1}\Sigma_{CY}$. $\square$

Unlike DR and its variants, whose variance can increase when the learned value function is misspecified, Shaped-PDIS admits a variance-minimizing control variate correction. In particular, optimizing $\lambda$ yields an estimator that is guaranteed not to increase variance relative to PDIS.

**Corollary 4.5** (**Bounded Variance of Shaped-PDIS**). *The optimal Shaped-PDIS estimator never has larger variance than PDIS*

$$\mathrm{Var}\big[\widehat{V}_{\mathrm{Shaped-PDIS}}(\lambda^\star)\big] \le \tfrac{1}{N}\mathrm{Var}[Y],$$

*with strict inequality iff* $\Sigma_{CY} \ne 0$.

Note that when $\lambda = -\mathbf{1} \in \mathbb{R}^{T+1}$, we recover a SCOPE-style shaping correction from Parbhoo et al. (2020) corresponding to uniformly aggregating the shaping feature columns. The resulting variance may be larger or smaller than PDIS depending on the alignment between $Y$ and the aggregated shaping features.

### 4.3. Learning control variates $\widetilde{C}^\Phi$ and $\lambda^*$ for variance guarantees

Theorem 4.4 shows that variance reduction is governed by the alignment between shaping features $\widetilde{C}^\Phi$ and PDIS return $Y$. Specifically, variance drops whenever $\widetilde{C}^\Phi$ correlates with IS-weighted returns $Y$. We therefore parameterize the potential function $\Phi_\beta : \mathcal{S} \to \mathbb{R}$ and learn $\beta$ from logged trajectories. This aligns the control variate with the return signal and thus guarantees variance reduction in OPE. The full procedure appears in Algorithm 1.

**Computing control variates $\widetilde{C}^\Phi$ using a potential network.** We parameterize $\Phi$ by a function approximator $\Phi_\beta$ (e.g., a linear model or neural network) and optimize its parameters $\beta$ for variance reduction. Given a trajectory

$\tau^{(n)} = (s_0^{(n)}, a_0^{(n)}, r_0^{(n)}, \ldots, s_T^{(n)})$, we compute a trajectory-level shaping feature vector $C^{\Phi_\beta,(n)} \in \mathbb{R}^{T+1}$

$$C^{\Phi_\beta,(n)} = \big(C_{-1}^{\Phi_\beta,(n)}, C_0^{\Phi_\beta,(n)}, \ldots, C_{T-1}^{\Phi_\beta,(n)}\big)^\top,$$

with components $C_{-1}^{\Phi_\beta,(n)} := \Phi_\beta(s_0^{(n)})$ and for $t = 0, \ldots, T-1$:

$$C_t^{\Phi_\beta,(n)} := \gamma^t W_t^{(n)}\Big(\gamma\Phi_\beta(s_{t+1}^{(n)}) - \Phi_\beta(s_t^{(n)})\Big).$$

Stacking these vectors and centering the features across trajectories yields,

$$\widetilde{C}^{\Phi_\beta} := C^{\Phi_\beta} - \mathbf{1}\,\mu_{C_\beta}^\top, \qquad \mu_{C_\beta} := \frac{1}{N}\sum_{n=1}^N C^{\Phi_\beta,(n)},$$

so that the resulting control variate features have zero empirical mean. Theorem 4.4 shows that, for a fixed choice of shaping features $\widetilde{C}^{\Phi_\beta}$, the variance-minimizing correction is obtained by choosing

$$\lambda^\star(\beta) = -\Sigma_{CC}(\beta)^{-1}\Sigma_{CY}(\beta).$$

**Optimizing the control variates $C^\Phi$ for variance reduction.** Substituting $\lambda^*(\beta)$ from above into the variance expression yields the exact variance reduction,

$$\begin{aligned}\mathrm{Var}(Y) &- \mathrm{Var}\big(Y + \lambda^\star(\beta)^\top\widetilde{C}^{\Phi_\beta}\big) \\ &= \Sigma_{CY}(\beta)^\top\Sigma_{CC}(\beta)^{-1}\Sigma_{CY}(\beta) \qquad (9)\end{aligned}$$

This motivates learning the potential parameters $\beta$ to maximize the explained-variance objective

$$J(\beta) := \widehat{\Sigma}_{CY}(\beta)^\top\widehat{\Sigma}_{CC}(\beta)^{-1}\widehat{\Sigma}_{CY}(\beta),$$

where $\widehat{\Sigma}_{CC}(\beta)$ and $\widehat{\Sigma}_{CY}(\beta)$ are computed empirically from the centered feature matrix $\widetilde{C}(\beta)$ and centered return vector $\widetilde{Y}$, optionally using ridge regularization

$$\widehat{\Sigma}_{\mathbf{CC}}(\beta) \leftarrow \widehat{\Sigma}_{\mathbf{CC}}(\beta) + \alpha I.$$

Note that in practice when $N, T$ are large, computing the matrix inversion in Eqn 9 becomes intractable. To reduce computational overload, one can resort to conjugate gradients (Shewchuk et al., 1994) (See Appendix D for details). Maximizing $J(\beta)$ directly optimizes the guaranteed variance-reduction term from Theorem 4.4, yielding shaping features that are both predictive of the return $Y$ and well-conditioned for stable coefficient estimation. Unlike standard supervised learning, there are no external labels: the only training signal comes from how well the shaping columns correlate with the return.

**Algorithm 1** Shaping-based OPE with learned Potential $\Phi_\beta$

---

1: **Input:** Trajectories $\{\tau^n\}$ from $\pi_b$, evaluation policy $\pi_e$, discount $\gamma$, potential model $\Phi_\beta$, folds $K$, (optional ridge $\alpha$).
2: **for** $k = 1$ **to** $K$ **do**
3:   Split data: fit set $\xi_{\text{fit}}$, evaluation set $\xi_k$.
4:   On $\xi_{\text{fit}}$, build $Y, \widetilde{C}^{\Phi_\beta}$ from IS returns $Y^{(n)}$ and shaping columns $\tilde{C}^{\Phi_\beta,(n)}$.
5:   Form covariances $\mathbf{\Sigma}_{CC}, \mathbf{\Sigma}_{CY}$
6:   Maximize $J(\beta) = \mathbf{\Sigma}_{CY}^\top \mathbf{\Sigma}_{CC}^{-1} \mathbf{\Sigma}_{CY}$ by backprop.
7:   Compute $\lambda_k = -\mathbf{\Sigma}_{CC}^{-1} \mathbf{\Sigma}_{CY}$ at $\hat{\beta}$.
8:   Evaluate on fold $\xi_k$: $\widehat{V}_k^{\pi_e} = \frac{1}{|\xi_k|} \sum_{n \in \xi_k} Y^{(n)} + \lambda_k^\top \tilde{C}^{\Phi_\beta,(n)}$.
9: **end for**
10: **Output:** $\widehat{V}_{\text{Shaped}-\text{PDIS}}^{\pi_e} = \frac{1}{K} \sum_k \widehat{V}_k^{\pi_e}$.

---

**Cross-fitting and the final estimator.** To prevent overfitting, we apply cross-fitting: the dataset is partitioned into $K$ folds, $\Phi_\beta$ and $\lambda^\star$ are learned on $K-1$ folds, and the estimator is evaluated on the held-out fold. For each fold $k$, the value estimate is

$$\widehat{V}_k^{\pi_e} = \frac{1}{I_k} \sum_{n \in I_k} \left( Y^{(n)} + \lambda^*(\widehat{\beta})^\top \tilde{C}^{\Phi_{\hat{\beta}},(n)} \right). \quad (10)$$

Finally, the overall estimate aggregates across folds,

$$\widehat{V}_{\text{Shaped}-\text{PDIS}}^{\pi_e}(\lambda) = \frac{1}{K} \sum_k \widehat{V}_k^{\pi_e}, \quad (11)$$

with empirical residual variance $\widehat{S}^2 = \frac{1}{K} \sum_k S_k^2$.

### 4.4. Relation to DR and MRDR

It is useful to contrast shaping-based estimators with the doubly robust (DR) and more robust doubly robust (MRDR) estimators. Both DR and MRDR employ the idea of control variates, but they rely on learning an approximate $Q$ function. This is fundamentally more challenging than learning a potential $\Phi_\beta$: the $Q$ function is state-action dependent, must satisfy a Bellman consistency condition, and directly encodes long horizon returns. By contrast, shaped OPE estimators requires only a scalar potential over states, grounded at terminal states. This yields 3 key advantages:

**Ease of learning.** While DR remains unbiased for *any* approximate $Q$, variance reduction depends critically on $Q$'s accuracy: a poorly estimated $Q$ can *increase* variance relative to plain IS. Under sparse rewards, learning an accurate $Q$ is fundamentally difficult as TD targets are mostly zero, Bellman consistency must hold across long horizons, and the function must generalize over state-action pairs. By contrast, $\Phi$ need not predict returns or satisfy any consistency condition; it only needs to correlate with the fluctuations in

the base estimator's return (whether PDIS, DR, or MRDR) so that adding $\lambda^\top \widetilde{C}^\Phi$ cancels out some of the variance. This simpler learning target can be optimized directly via $J(\beta)$ using dense state visitations, even when reward signal is nearly absent.

**Interpretability.** Our approach is more interpretable than DR-style estimators, which rely on a learned action-value model $Q(s,a)$ whose errors can be difficult to diagnose and may introduce unpredictable variance behavior. In contrast, Shaped-PDIS uses a state-based potential $\Phi(s)$ that induces transparent shaping features along the observed trajectory, making it easier to attribute variance reduction to specific regions of the state space and timesteps. Learned coefficients $\lambda$ further provide a direct measure of which shaping components are most influential. This yields a lightweight, auditable variance-reduction mechanism that complements OPE without requiring a full reward-model or $Q$-function.

**Tighter Concentration Bounds.** Bernstein-type concentration bounds for OPE scale with the estimator variance. For Shaped-PDIS, the per-trajectory quantity that defines this variance is $Y + \lambda^\top \widetilde{C}^\Phi$. Thus for any fixed $\lambda$ and $\delta \in (0,1)$,

$$\left| \hat{V}_{\text{Shaped-PDIS}}^{\pi_e} - V^{\pi_e} \right| \leq \sqrt{\frac{2 \operatorname{Var}(Y + \lambda^\top \widetilde{C}^\Phi) \log(2/\delta)}{N}} + O\left( \frac{\log(1/\delta)}{N} \right).$$

At the projection-optimal coefficients $\lambda^\star$, Theorem 4.4 guarantees the monotone variance bound,

$$\operatorname{Var}(Y + \lambda^{\star\top} \widetilde{C}^\Phi) \leq \operatorname{Var}(Y).$$

Therefore Shaped-PDIS yields uniformly tighter variance-controlled concentration than PDIS. DR-style estimators rely on a learned model-based control variate of $Q$ whose misspecification can increase the estimator variance, and therefore do not admit analogous tightening guarantees of Bernstein-type bounds relative to PDIS. Analogous bounds for shaped variants of DR/MRDR appear in Appendix C.1.

### 4.5. Finite-sample guarantees

Until now, we assumed oracle access to $\mathbf{\Sigma}_{CC}, \mathbf{\Sigma}_{CY}$, and $\mu_C$. In practice, all three are estimated from $\mathcal{D}_{\text{fit}}$ which composes of finite samples. $\Phi_\beta$ is estimated parametrically, and importance weights are bounded only by $\rho_{\max}^T$. We provide a complete non-asymptotic analysis in Appendix D, with three takeaways worth highlighting here. First, cross-fitting yields *exact* finite-sample unbiasedness for any data-dependent $\widehat{\lambda}^{\text{fit}}$ (Theorem D.1), strengthening the population statement of Theorem 4.3. Second, validity is structurally decoupled from estimation quality: the boundary condition $\Phi(s_T) = 0$ guarantees zero-mean shaping features regardless of how well $\Phi_\beta$ generalizes, so misspecification of $\mathcal{B}$,

optimization failures, and finite-sample noise affect only the magnitude of variance reduction, not the validity of the estimator (Theorem D.8). Third, the population variance reduction $\boldsymbol{\Sigma}_{CY}^{\top} \boldsymbol{\Sigma}_{CC}^{-1} \boldsymbol{\Sigma}_{CY}$ must dominate a plug-in penalty of order $\kappa(\boldsymbol{\Sigma}_{CC} + \alpha I)^2 \cdot T/N_1$ for the shaped estimator to beat its unshaped counterpart in finite samples (Corollary D.6). The same analysis extends to Shaped-DR and Shaped-MRDR by replacing $Y$ with $Y_{\mathrm{DR}}$ or $Y_{\mathrm{MRDR}}$.

# 5. Experiments

Our experiments stress-test OPE estimators across domains where standard methods fail: long horizons, sparse rewards, noisy signals, and real-world clinical trajectories. We evaluate on 11 environments: a tabular Chain MDP, a cancer simulator, Pointmaze, Antmaze, 5 single-stock and 1 multi-stock DOW-30 trading task, and an ICU-Sepsis benchmark. We parameterize $\Phi_\beta$ as a two-layer tanh MLP (architecture varies by domain; see Appendix F), trained via Adam to maximize $J(\beta)$ with ridge regularization for stable covariance inversion. Optimal coefficients $\lambda^*$ are computed via Theorem 4.4 using 5-fold cross-fitting. Full hyperparameter details in Appendix F.

**Baselines.** We compare against PDIS, DR (with FQE critic), and MRDR (importance-weighted regression critic). Each shaped variant adds reward-shaping control variates, with $\Phi_\beta$ learned by maximizing explained variance between base returns and shaping features, ensuring unbiased control variates directly optimized for variance reduction.

**Metrics.** We report bias, variance, MSE and ESS. Detailed metric descriptions in Appendix E.1, experimental setups in Appendix F. Experiments were performed over 10 seeds.

## 5.1. Evaluation on DOW-30 Trading Benchmark

**Environment Structure.** We consider 5 single-stock and 1 multi-stock trading task on DOW-30 stocks following (Liu et al., 2022) in 2025-26. Trajectories span 253 trading days with a starting budget of $100k. States comprise current budget, shares held, and closing prices: $s_t \in \mathbb{R}^3$ for single-stock and $s_t \in \mathbb{R}^{90}$ for multi-stock. Actions are buy 50%, hold, or sell 50% per ticker. The reward is final minus initial portfolio value, with no intermediate rewards. Large state-action spaces, long horizons, and daily price fluctuations make this an extremely challenging benchmark.

**Policies.** We train PPO for 25k trajectories to obtain an optimal policy. The evaluation policy $\pi_e$ and behavior policy $\pi_b$ are 10% and 15% $\epsilon$-greedy variants, respectively. Ground-truth $V^{\pi_e}$ is estimated via Monte Carlo rollouts.

## 5.2. Evaluation on ICU-Sepsis Benchmark

**Environment Structure.** We evaluate on the ICU-Sepsis benchmark (Choudhary et al., 2024) derived from real-world

MIMIC-III. Patient trajectories are segmented into 4-hour windows over horizons up to 72 hours. Each state $s_t$ has 47 features including vital signs, labs, demographics, and derived scores (e.g., SOFA), all normalized and imputed per the benchmark. Actions $a_t$ are intravenous fluids and vasopressors, discretized into a $5 \times 5$ grid (25 actions). Rewards are sparse: $+1$ if discharged alive, 0 otherwise, reflecting the clinical challenge of evaluating with delayed outcomes.

**Policies.** We train PPO for 1M episodes, taking $\pi_b$ as the actor at episode 250k and $\pi_e$ at episode 1M; their optimality gap induces distribution shift. Ground-truth $V^{\pi_e}$ is estimated via Monte Carlo rollouts.

# 6. Results and Discussion

## 6.1. Quantitative Analysis

**Shaped estimators have orders of improvement in variance, MSE and ESS while being unbiased over traditional estimators.** We compare shaped and traditional OPE estimators across all 11 environments. Figures 2 and 10 (Appendix G) present results for ICU-Sepsis and the other benchmarks. Across all environments, shaped estimators consistently outperform their traditional counterparts on every metric. The improvements are particularly striking for ICU-Sepsis, where variance and MSE of the shaped estimators are 3-4 orders of magnitude lower than the traditional estimators. Consequently, the ESS of shaped estimators is substantially higher than the traditional variants. Consistent with our theoretical findings, all shaped estimators remain asymptotically unbiased.

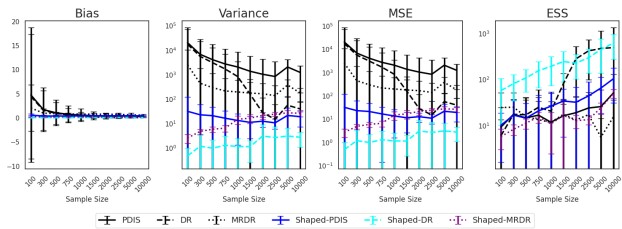

*Figure 2.* ICU-Sepsis. We observe the variance, MSE of the shaped estimators is lower than the traditional estimators, while being unbiased. The ESS of the shaped-estimators has higher mean and lower deviation as compared to the shaped-estimators.

**Shaped estimators are more robust to noise than traditional OPE estimators.** Figures 3 and 12 (Appendix G) present results under varying levels of Gaussian noise added to the final returns for ICU-Sepsis and DOW-30 trading environments respectively. Shaped estimators remain consistently more robust than their traditional counterparts as noise increases. Traditional estimators, by contrast, rely heavily on these sparse signals, so any corruption disproportionately degrades performance.

**Shaped estimators have superior performance in sparse**

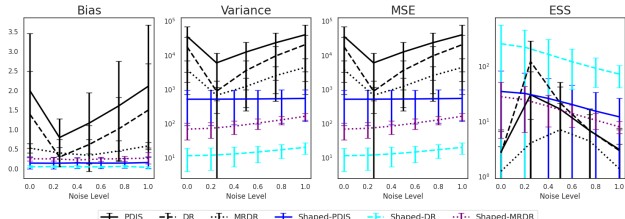

*Figure 3.* ICU-Sepsis Noise comparison. We observe the shaped estimators are more robust to traditional estimators when the reward signal is contaminated with gaussian noise.

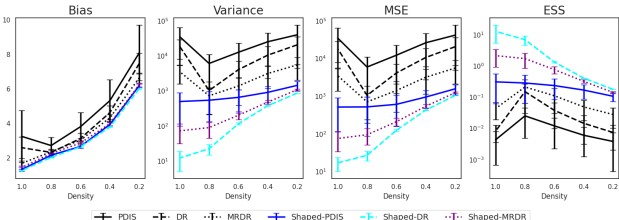

*Figure 4.* ICU-Sepsis Density comparison. We observe the shaped estimators are superior to traditional estimators when the reward signals are sparser. Experiments were conducted over 10 seeds.

**reward settings.** Figures 4 and 13 (Appendix G) present results under varying reward sparsity for ICU-Sepsis and DOW-30 stocks. For ICU-Sepsis, we vary sparsity by removing the final transition (containing the return) from $(1 - d)\%$ of trajectories, where $d$ is the density parameter. For DOW-30, we define density as the number of timesteps that have an intermediate reward, with 0.0 indicating only terminal reward and 1.0 indicating intermediate rewards across all timesteps. Under both settings, shaped estimators consistently outperform their non-shaped counterparts and exhibit slower performance deterioration as sparsity increases, confirming their ability to extract signal from both successful and unsuccessful trajectories. Among shaped variants, Shaped-DR and Shaped-MRDR outperform Shaped-PDIS, demonstrating the complementary benefits of combining shaping-based control variates with model-based corrections.

### 6.2. Qualitative Analysis

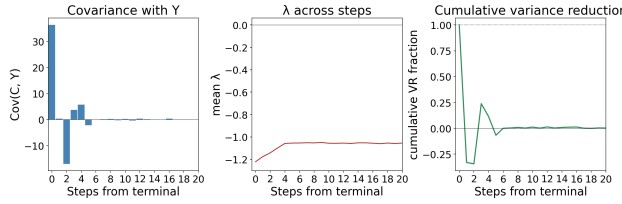

*Figure 5.* ICU-Sepsis Variance Analysis. We observe, maximum variance reduction happens in the last few timesteps of the trajectory, indicated by high covariance between $C_t$ and $Y_{\text{PDIS}}$, and 0 variance reduction throughout the trajectory and 100% variance reduction in the last 5 timesteps.

**Most of the variance reduction through shaping happens**

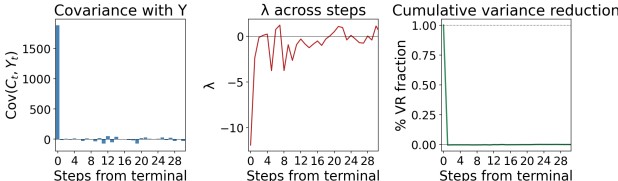

*Figure 6.* DOW-30 Variance Analysis on Apple Stock. We observe variance reduction primarily occurs at the timestep $T - 1$ with $\lambda_t = 0$, and cumulative variance reduction is 0 for $t = 0$ to $T - 2$.

**at the later stages of the trajectory.** Figures 5 and 6 show that variance reduction for ICU-Sepsis and Apple concentrates at the penultimate step $(T-1)$ and the last five steps, respectively. This is surprising, since the optimizer has $T+1$ columns available yet uses essentially the reward-adjacent ones. The structure of the objective explains this.

The shaped estimator is $Y_{\text{shaped}} = Y + \sum_k \lambda_k \tilde{C}_k^\Phi$ with inner columns $\tilde{C}_t^\Phi = \gamma^t W_t(\gamma \Phi(s_{t+1}) - \Phi(s_t))$, and the optimal weighting attains $J(\lambda^\star(\beta)) = \Sigma_{CY}^\top \Sigma_{CC}^{-1} \Sigma_{CY}$. Thus $\Phi$ need not track $r_t$ pointwise; it must only build columns whose IS-weighted increments *co-vary with $Y$ across trajectories*, and the shaping concentrates wherever $\text{Cov}(\tilde{C}_k^\Phi, Y)$ can be made large relative to the redundancy in $\Sigma_{CC}$.

Two factors localize this covariance near the reward. First, in the sparse-reward regime $Y = \gamma^T W_T r_T$, so $Y$ carries the full cumulative weight $W_T = \prod_j \rho_j$. The late column $\tilde{C}_{T-1}^\Phi$ shares this same $W_T$ multiplicatively, so $|\text{Cov}(\tilde{C}_{T-1}^\Phi, Y)|$ is boosted by the shared importance-weight variation alone, before $\Phi$ adds any state signal. An early column $\tilde{C}_t^\Phi$ ($t \ll T$) carries only the partial weight $W_t$; to co-vary with $Y$ it would need $s_t$ to predict the entire downstream product $W_{T-1} r_{T-1}$, i.e. $\Phi$ must anticipate the future ratios $\rho_{t+1}, \ldots, \rho_{T-1}$ *and* the reward. The reward is readable from the late state, but the future ratios are not encoded in $s_t$, so the early-column covariance is weak.

Second, even though the early columns carry *some* usable signal, the objective gains almost nothing by adding them. Because $W_{k+1} = W_k \rho_{k+1}$, adjacent columns are built from nearly the same cumulative weight and are thus highly correlated, largely duplicating one another. The $\Sigma_{CC}^{-1}$ term accounts for exactly this overlap: redundant columns are credited only once, not additively. Spreading the covariance across a band of correlated early columns therefore does not accumulate into a larger $J$. The reward-adjacent column is the one piece of information its neighbors cannot reconstruct, because of the largest importance weight, hence $\Phi$ concentrates the variance reduction effort in the final timesteps.

**The shaping potential $\Phi$ and its control variate $\tilde{C}_t^\Phi$ are not value functions.** Figures 7 and 8 track, across a varied set of trajectories, the on-policy value $V^{\pi_e}(s_t)$ alongside the shaping potential $\Phi(s_t)$, the control-variate column $\tilde{C}_t^\Phi$, and

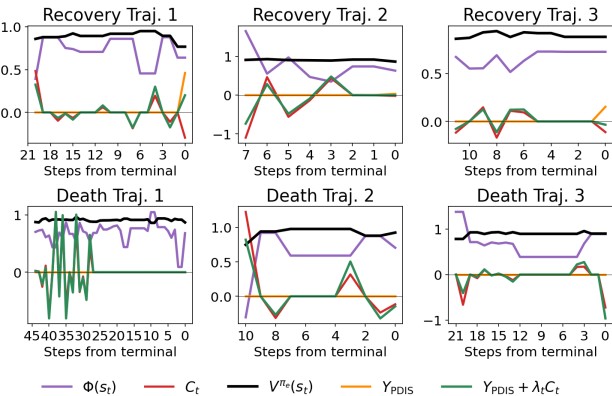

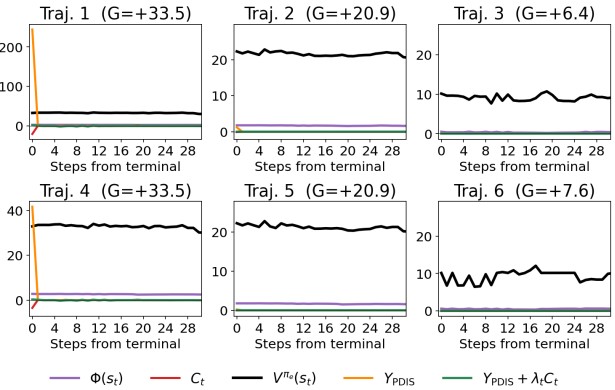

*Figure 7.* ICU-Sepsis: Decoupling of $\Phi$ from $V$. We consider 6 patient trajectories, with *(Top)* being recovery patients and *(Bottom)* being death patients. Across all trajectories, we notice the true on-policy value function is fundamentally different from the potential $\Phi$, cumulative per-step shaping $\tilde{C}_t^{\Phi}$, and the estimators $Y_{\text{PDIS}}$ and $Y_{\text{Shaped-PDIS}}$. This indicates our shaping algorithm primarily focuses on variance reduction and doesn't correlate with value functions that satisfy the Bellman equation.

*Figure 8.* DOW-30 Apple Stock: Decoupling of $\Phi$ from $V$. We demonstrate 6 trajectories, 2 leading to maximum profits (Column 1), 2 leading to average profits (Column 2), and 2 leading to minimum profits (Column 3). Similar to ICU-Sepsis, we observe there is no correlation between $V_{\pi_e}$ and $\tilde{C}_t^{\Phi}$, $\Phi$. This indicates $\tilde{C}_t^{\Phi} \neq V$ in general.

the estimators $Y_{\text{PDIS}}$ and $Y_{\text{Shaped-PDIS}}$. Across both domains and all trajectories these quantities show little correlation: $\Phi$ and $\tilde{C}_t^{\Phi}$ trace patterns that do not align with $V^{\pi_e}$ either in shape or in magnitude. This is expected. A value function is pinned down by the Bellman equation, $V^{\pi_e}(s_t)$ must equal the expected return-to-go from $s_t$ whereas $\Phi$ and $\tilde{C}_t^{\Phi}$ are optimized solely to reduce the variance of $Y_{\text{PDIS}}$ and are under no such constraint. $\Phi$ is free to take whatever shape makes the IS-weighted increments $\tilde{C}_t^{\Phi}$ co-vary with the return; the Bellman-consistent value function is merely one point in a far larger space of potentials, and almost never the one variance reduction selects. Hence $\Phi, \tilde{C}_t^{\Phi} \neq V$ in general, and the resemblance some shaping methods assume between learned potentials and value functions does not hold here.

**Shaping potentials could potentially reveal meaningful state structure.** While $\Phi$ is not a value function, in favorable regimes it can still encode interpretable state quality. The clearest case is Apple, Figure 9): $\Phi(s_{T-1})$ varies linearly with the terminal reward $r_T$ at a small slope ($\approx 0.08$). The slope and sign play distinct roles: the small coefficient scales every return down uniformly, producing the variance reduction, while the sign carries the meaning: because $\Phi$ moves in step with $r_T$, high-

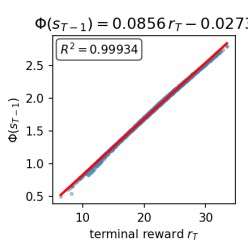

*Figure 9.* Apple $\Phi$ Qualitative Analysis. The shaping function $\Phi$ at $T-1$ varies linearly with the terminal reward. The slope ($\approx 0.08$) scales rewards down to reduce variance; qualitatively, $\Phi > 0$ marks a high-portfolio (good) state and $\Phi < 0$ a bad one, learnt through $\Phi$-shaping.

portfolio states receive $\Phi > 0$ and poor ones $\Phi < 0$. The potential thus orders states by quality without ever being trained to, since here the variance reduction requires it to encode the terminal reward which happens to coincide with state quality. This does not generalize. Apple is favorable because a single state is shaped and the importance weights are mild, leaving $\Phi$ free to align with reward magnitude. In ICU-Sepsis many states are shaped and variance reduction is dominated by cancelling importance-weight dispersion rather than outcome variation; $\Phi$ is consumed entirely by this and cannot also serve as an interpretable score. Interpretability survives only when IS-weighting don't compete for $\Phi$'s limited capacity. We leave denser shaping schemes that recover both variance reduction *and* qualitative structure to future work.

## 7. Conclusions and Future Work

This paper introduced reward shaping-based control variates for OPE in sparse and noisy reward settings. We derived a family of shaping-based estimators and showed theoretically and empirically that these estimators can both reduce variance and preserve the policy value, while remaining unbiased, even under the presence of unsuccessful sparse reward trajectories. Notably, the shaped OPE estimators offer significant advantages in terms of their interpretability as their values directly rank helpful and harmful states and reveal the features that drive them, offering insights that are harder to obtain from DR methods. Future work should explore how shaping-based OPE methods perform in partially observable environments, as well as hybrid shaping-based and marginalized estimators, and prospective studies where reward shaping control variates guide offline policy selection and safe deployment.

## Impact Statement

This work advances OPE in sparse-reward settings common to healthcare, finance, and other high-stakes domains where deploying untested policies carries significant risk. A key advantage of our approach is that it extracts useful signal from all logged trajectories, including the majority that fail, rather than relying solely on rare successes, enabling reliable evaluation even when logged data contains almost no reward information. By enabling more robust policy evaluation from limited data, our methods could help practitioners identify effective treatment strategies or decision policies without costly and potentially harmful online experimentation.

## Acknowledgements

The authors would like to thank Omer Gottesman for feedback on early versions of this work, and Elizaveta Sheremetyeva for advice on improving the presentation quality of this work.

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

## A. Preliminaries: Existing OPE estimators

The goal of our approach is to reduce the variance and MSE of off-policy evaluation in sparse reward settings. Instead of using a value function as a control variate like DR methods, we use a potential-based reward shaping control variate and optimize over the space of shaped rewards. Because of its resemblance to DR, we provide a brief overview of the IS and DR approaches.

**Per-Decision Importance Sampling.** Importance sampling (IS) estimators reweight observed trajectories to correct for the distribution mismatch between $\pi_b$ and $\pi_e$. The standard trajectory-wise IS estimator is given by

$$\hat{V}_{\text{IS}}^{\pi_e} = \frac{1}{N} \sum_{n=1}^{N} W_{T-1}^{(n)} R_{0:T-1}(\tau^{(n)}); \qquad W_{T-1}^{(n)} = \prod_{k=0}^{T-1} \frac{\pi_e(a_k \mid s_k)}{\pi_b(a_k \mid s_k)}, \tag{12}$$

is the importance ratio of the trajectory.

This estimator is unbiased but suffers from exponential variance in horizon length. To address this, the *per-decision IS (PDIS)* estimator decomposes the return into stepwise contributions:

$$\widehat{V}_{\text{PDIS}}^{\pi_e} = \frac{1}{N} \sum_{n=1}^{N} \sum_{t=0}^{T-1} \gamma^t W_t^{(n)} r_t^{(n)}, \qquad W_t^{(n)} = \prod_{k=0}^{t} \frac{\pi_e(a_k^{(n)} \mid s_k^{(n)})}{\pi_b(a_k^{(n)} \mid s_k^{(n)})}, \tag{13}$$

Using partial trajectory overlap, PDIS achieves lower variance than trajectory-wise IS, but still relies on frequent reward observations and performs poorly in sparse-reward settings.

**Doubly Robust Off-Policy Evaluation.** DR estimators combine DM with IS and have been widely used in regression, contextual bandits (Dudik et al., 2011), and RL (e.g. Thomas & Brunskill (2016); Jiang & Li (2016)). In RL, the DR estimate is given by,

$$\hat{V}_{\text{DR}}^{\pi_e}(\beta) = \frac{1}{N} \sum_{n=1}^{N} \sum_{t=0}^{T-1} \gamma^t \left[ W_t^{(n)} r_t^{(n)} - W_t^{(n)} \hat{Q}^{\pi_e}(s_t^{(n)}, a_t^{(n)}; \beta) + W_{t-1}^{(n)} \hat{V}^{\pi_e}(s_t^{(n)}; \beta) \right]. \tag{14}$$

The IS part of DR is based on step-IS while the model part relies on $\hat{Q}^{\pi_e}$ and $\hat{V}^{\pi_e}$ model estimates. Importantly, the bias of the DR estimator is a product of both the bias of DM and IS. As a result, DR is unbiased if either IS or DM is unbiased. When the behaviour policy $\pi_b$ is known, Eq. 14 is unbiased. The MRDR estimator (Farajtabar et al., 2018) modifies classic DR by learning the model parameter that minimizes the variance of the DR estimator.

## B. A Family of Shaped Estimators and Their Properties

In addition to the standard Shaped PDIS estimator we present in the main paper, the idea of reward shaping control variates can be integrated into existing OPE estimators with similar performance guarantees in sparse reward settings. Here, we show in particular how shaped control variates can be integrated into DR and MRDR estimators.

**Definition B.1** (**Shaped DR estimator**). Given $N$ i.i.d. trajectories, the shaped DR estimator augments DR with a reward shaping control variate $C^\Phi$,

$$\widehat{V}_{\text{Shaped}-\text{DR}}^{\pi_e}(\lambda) := \widehat{V}_{\text{DR}}^{\pi_e} + \lambda^\top \left( \frac{1}{N} \sum_{n=1}^{N} \widetilde{C}^{\Phi,(n)} \right), \tag{15}$$

where $\lambda \in \mathbb{R}^{T+1}$ is a coefficient vector controlling the weight of the centered shaping feature vector (as in Shaped-PDIS).

**Theorem B.2** (**Bias of Shaped DR**). *Under Assumptions, Lemma 4.2, and standard DR conditions,*

$$\text{Bias}\left( \widehat{V}_{\text{Shaped-DR}}^{\pi_e}(\lambda) \right) := \mathbb{E}_{\pi_b}\left[ \widehat{V}_{\text{Shaped-DR}}^{\pi_e}(\lambda) \right] - V^{\pi_e} \tag{16}$$

$$= \mathbb{E}_{\pi_b}[\widehat{V}_{\text{DR}}^{\pi_e}] + \lambda^\top \mathbb{E}_{\pi_b}\left[ \frac{1}{N} \sum_{n=1}^{N} \widetilde{C}^{\Phi,(n)} \right] - V^{\pi_e} \tag{17}$$

$$= V^{\pi_e} + \lambda^\top \mathbb{E}_{\pi_b}[\widetilde{C}^\Phi] - V^{\pi_e} \tag{18}$$

$$= 0, \tag{19}$$

*since $\widetilde{C}^\Phi$ are zero-mean control variates. Hence the Shaped-DR estimator is an unbiased estimator of $V^{\pi_e}$.*

**Theorem B.3** (**Variance of Shaped DR**). *The variance of the Shaped DR estimator is given by,*

$$\text{Var}(\widehat{V}^{\pi_e}_{\text{Shaped}-\text{DR}}(\lambda)) = \frac{1}{N}\text{Var}(Y_{DR}) + 2\lambda^\top \boldsymbol{\Sigma}^{\text{DR}}_{CY} + \lambda^\top \boldsymbol{\Sigma}_{CC}\lambda \tag{20}$$

*This variance is optimized at $\lambda^* = -\boldsymbol{\Sigma}^{-1}_{CC}\boldsymbol{\Sigma}^{\text{DR}}_{CY}$.*

*Proof.* The proof follows directly from the proof of Theorem 4.4, only replacing $Y$ with the DR-based return, $Y_{\text{DR}}$ and $\boldsymbol{\Sigma}_{CY}$ with its DR-counterpart, $\boldsymbol{\Sigma}^{\text{DR}}_{CY}$. □

**Definition B.4** (**Shaped MRDR estimator**). Given $N$ i.i.d. trajectories, the shaped MRDR estimator augments MRDR with a reward shaping control variate $C^\Phi$:

$$\widehat{V}^{\pi_e}_{\text{Shaped}-\text{MRDR}}(\lambda) := \widehat{V}^{\pi_e}_{\text{MRDR}} + \lambda^\top \Big(\frac{1}{N}\sum_{n=1}^N \widetilde{C}^{\Phi,(n)}\Big), \tag{21}$$

where $\lambda \in \mathbb{R}^{T+1}$ is a coefficient controlling the weight of the zero-mean control variate applied.

**Theorem B.5** (**Bias of Shaped-MRDR**). *Under Assumptions, Lemma 4.2, and MRDR unbiasedness, $\widehat{V}_{\text{Shaped-MRDR}}(\lambda)$ is unbiased:*

$$\mathbb{E}[\widehat{V}_{\text{Shaped-MRDR}}(\lambda)] = V^{\pi_e}.$$

*Proof.* MRDR is an instance of DR with variance-minimizing models. Hence by the same reasoning as Theorem B.2, expectation equals $V^{\pi_e}$. □

**Theorem B.6** (**Variance of Shaped MRDR**). *The variance of the Shaped MRDR estimator is given by,*

$$\text{Var}(\widehat{V}^{\pi_e}_{\text{Shaped}-\text{MRDR}}(\lambda)) = \frac{1}{N}\text{Var}(Y_{MRDR}) + 2\lambda^\top \boldsymbol{\Sigma}^{\text{MRDR}}_{CY} + \lambda^\top \boldsymbol{\Sigma}_{CC}\lambda \tag{22}$$

*This variance is optimized at $\lambda^* = -\boldsymbol{\Sigma}^{-1}_{CC}\boldsymbol{\Sigma}^{\text{MRDR}}_{CY}$.*

*Proof.* The proof follows directly from the proof of Theorem 4.4, only replacing $Y$ with the MRDR-based return, $Y_{\text{MRDR}}$ and $\boldsymbol{\Sigma}_{CY}$ with its MRDR-counterpart, $\boldsymbol{\Sigma}^{\text{MRDR}}_{CY}$. □

## C. Reward Shaping Control Variates for Variance Reduction

As shown in Section 4, for any choice of base statistic $Y$, the variance of the shaped per-trajectory quantity $Y + \lambda^\top \widetilde{C}_\Phi$ is a quadratic function of $\lambda$.

Define the covariance objects

$$\boldsymbol{\Sigma}_{CC} := \text{Cov}_{\pi_b}(\widetilde{C}^\Phi, \widetilde{C}^\Phi) \in \mathbb{R}^{(T+1)\times(T+1)}, \qquad \boldsymbol{\Sigma}_{CY} := \text{Cov}_{\pi_b}(\widetilde{C}^\Phi, Y) \in \mathbb{R}^{T+1}.$$

Then the variance-minimizing coefficient vector is given by

$$\lambda^\star = -\boldsymbol{\Sigma}^{-1}_{CC}\boldsymbol{\Sigma}_{CY}.$$

Equivalently, $\lambda^\star$ is the solution to the linear system

$$\boldsymbol{\Sigma}_{CC}\lambda = -\boldsymbol{\Sigma}_{CY}. \tag{23}$$

**Conjugate gradients for computing $\lambda^\star$.** Rather than explicitly forming $\Sigma_{CC}^{-1}$, we compute $\lambda^\star$ by solving (23) using the conjugate gradient method. This is appropriate since $\Sigma_{CC}$ is symmetric positive semidefinite by construction (as a covariance matrix). Conjugate gradients requires only matrix-vector products $v \mapsto \Sigma_{CC}v$, where $v \in \mathbb{R}^{T+1}$ is an arbitrary trial/search vector during iterative solve.

In practice, we estimate $\Sigma_{CC}$ and $\Sigma_{CY}$ from data as

$$\widehat{\Sigma}_{CC} := \frac{1}{N} \sum_{n=1}^{N} \widetilde{C}^{\Phi,(n)} \left( \widetilde{C}^{\Phi,(n)} \right)^\top ,$$

$$\widehat{\Sigma}_{CY} := \frac{1}{N} \sum_{n=1}^{N} \widetilde{C}^{\Phi,(n)} Y^{(n)}.$$

Each conjugate gradient iteration requires computing $\widehat{\Sigma}_{CC}v$, which can be implemented without materializing $\widehat{\Sigma}_{CC}$ explicitly via

$$\widehat{\Sigma}_{CC}v = \frac{1}{N} \sum_{n=1}^{N} \widetilde{C}^{\Phi,(n)} \left( \{\widetilde{C}^{\Phi,(n)}\}^\top v \right).$$

This yields an $O(N(T+1))$ cost per iteration.

**Regularization.** To ensure numerical stability when $\widehat{\Sigma}_{CC}$ is ill-conditioned or low-rank, we solve the regularized system

$$(\widehat{\Sigma}_{CC} + \eta I)\lambda = -\widehat{\Sigma}_{CY}$$

for a small $\eta > 0$.

### C.1. Confidence Bounds of Shaped Estimators

In Section 4, we provided theoretical bounds on the performance of Shaped-PDIS, demonstrating that these performance bounds were tighter than those of standard PDIS. However, since standard OPE estimators often have vacuous high-probability bounds, in this section, we provide tighter bounds for Shaped-DR and Shaped-MRDR estimators in comparison to their non-shaped counterparts.

**Confidence bounds of Shaped DR Estimator.** Assume boundedness (for some $R, B, w > 0$) ensuring a uniform envelope

$$|Y_{\mathrm{DR}} + \lambda^\top \widetilde{C}^\Phi| \leq L_{\mathrm{DR}}(\lambda), \qquad \forall \lambda \in \mathbb{R}^{T+1}. \tag{24}$$

Let

$$\sigma_{\mathrm{DR}}^2(\lambda) := Var[Y_{\mathrm{DR}} + \lambda^\top \widetilde{C}^\Phi]. \tag{25}$$

Then, for any $\delta \in (0, 1)$, Bernstein's inequality yields

$$\left| \widehat{V}_{\text{Shaped-DR}}(\lambda) - V^{\pi_e} \right| \leq \sqrt{\frac{2\,\sigma_{\mathrm{DR}}^2(\lambda)}{N} \log\frac{2}{\delta}} + \frac{2\,L_{\mathrm{DR}}(\lambda)}{3N} \log\frac{2}{\delta}. \tag{26}$$

At $\lambda = \lambda_{\mathrm{DR}}^\star$, we have $\sigma_{\mathrm{DR}}^2(\lambda_{\mathrm{DR}}^\star) \leq Var[Y_{\mathrm{DR}}]$, so the Bernstein CI for Shaped–DR is (weakly) tighter than that for DR at the same confidence.

Let $S_N^2(\lambda)$ denote the sample variance of $\{Y_{\mathrm{DR},i} + \lambda^\top \widetilde{C}_i^\Phi\}_{i=1}^N$. The empirical Bernstein bound gives

$$\left| \widehat{V}_{\text{Shaped-DR}}(\lambda) - V^{\pi_e} \right| \leq \sqrt{\frac{2\,S_N^2(\lambda)}{N} \log\frac{3}{\delta}} + \frac{3\,L_{\mathrm{DR}}(\lambda)}{N} \log\frac{3}{\delta}. \tag{27}$$

At $\lambda_{\mathrm{DR}}^\star$, $S_N^2(\lambda_{\mathrm{DR}}^\star)$ concentrates below the DR sample variance, so the empirical intervals are likewise tighter.

**Confidence bounds of Shaped MRDR Estimator.** Assume a uniform envelope

$$|Y_{\text{MRDR}} + \lambda^\top \widetilde{C}^\Phi| \leq L_{\text{MRDR}}(\lambda), \qquad \forall \lambda \in \mathbb{R}^{T+1}. \tag{28}$$

Let

$$\sigma^2_{\text{MRDR}}(\lambda) := Var[Y_{\text{MRDR}} + \lambda^\top \widetilde{C}^\Phi]. \tag{29}$$

Then, for any $\delta \in (0, 1)$,

$$\left|\widehat{V}_{\text{Shaped-MRDR}}(\lambda) - V^{\pi_e}\right| \leq \sqrt{\frac{2\,\sigma^2_{\text{MRDR}}(\lambda)}{N} \log\frac{2}{\delta}} + \frac{2\,L_{\text{MRDR}}(\lambda)}{3N} \log\frac{2}{\delta}. \tag{30}$$

At $\lambda^\star_{\text{MRDR}}$, $\sigma^2_{\text{MRDR}}(\lambda^\star_{\text{MRDR}}) \leq Var[Y_{\text{MRDR}}]$, so the Bernstein CI for Shaped–MRDR is (weakly) tighter than that for MRDR.

Let $S^2_N(\lambda)$ be the sample variance of $\{Y_{\text{MRDR},i} + \lambda^\top \widetilde{C}_i^\Phi\}_{i=1}^N$. The empirical Bernstein inequality implies

$$\left|\widehat{V}_{\text{Shaped-MRDR}}(\lambda) - V^{\pi_e}\right| \leq \sqrt{\frac{2\,S^2_N(\lambda)}{N} \log\frac{3}{\delta}} + \frac{3\,L_{\text{MRDR}}(\lambda)}{N} \log\frac{3}{\delta}. \tag{31}$$

Evaluated at $\lambda^\star_{\text{MRDR}}$, these intervals are tighter.

## D. Finite-Sample Analysis of Shaped OPE Estimators

The guarantees in Section 4 are population-level: they characterize $\lambda^\star$ and the corresponding variance reduction $\Sigma_{CY}^\top \Sigma_{CC}^{-1} \Sigma_{CY}$ assuming exact knowledge of $\Sigma_{CC}, \Sigma_{CY}$, and $\mu_C$. In practice all three are estimated from finitely many trajectories, $\Phi_\beta$ is itself learned, and importance weights are unbounded. This appendix provides non-asymptotic bounds that quantify how each source of estimation error enters the final estimator, and identifies the regimes in which the shaped estimator provably improves on its unshaped counterpart.

We present results for Shaped-PDIS in Sections D.1–D.7, and extend them to Shaped-DR/MRDR in Section D.8. Throughout, $N$ denotes the total number of trajectories, $K$ the number of cross-fitting folds, $N_1 = N(K-1)/K$ the fit-fold size, and $N_2 = N/K$ the eval-fold size. We use $\widetilde{O}$ to suppress polylogarithmic factors.

### D.1. Setup and assumptions

For a fixed potential $\Phi : \mathcal{S} \to \mathbb{R}$ with $\Phi(s_T) = 0$, recall the per-trajectory quantities

$$Y(\tau) = \sum_{t=0}^{T-1} \gamma^t W_t r_t,$$

$$C^\Phi(\tau) = \left(\Phi(s_0),\, \gamma^0 W_0(\gamma\Phi(s_1) - \Phi(s_0)),\, \ldots,\, \gamma^{T-1} W_{T-1}(\gamma\Phi(s_T) - \Phi(s_{T-1}))\right)^\top \in \mathbb{R}^{T+1}.$$

Let $\mu_C = \mathbb{E}_{\pi_b}[C^\Phi]$, $\Sigma_{CC} = \text{Cov}_{\pi_b}(C^\Phi)$, $\Sigma_{CY} = \text{Cov}_{\pi_b}(C^\Phi, Y)$, and $\sigma_Y^2 = \text{Var}_{\pi_b}(Y)$. The shaped estimator is

$$\widehat{V}(\lambda) = \frac{1}{N_2} \sum_{n \in \mathcal{D}_{\text{eval}}} \left(Y^{(n)} + \lambda^\top \left(C^{\Phi,(n)} - \widehat{\mu}_C^{\text{fit}}\right)\right),$$

where $\widehat{\mu}_C^{\text{fit}}, \widehat{\Sigma}_{CC}^{\text{fit}}, \widehat{\Sigma}_{CY}^{\text{fit}}$, and $\widehat{\lambda}^{\text{fit}}$ are computed on $\mathcal{D}_{\text{fit}}$ (independent of the eval fold).

**Standing assumptions.**

(A1) *Bounded importance weights.* There exists $\rho_{\max} \geq 1$ such that $\sup_{s,a} \pi_e(a\,|\,s)/\pi_b(a\,|\,s) \leq \rho_{\max}$. Combined with $|r_t| \leq R_{\max}$ and $|\Phi| \leq B_\Phi$, this yields the deterministic envelopes

$$|Y| \leq L_Y := R_{\max}\rho_{\max}^T \frac{1-\gamma^T}{1-\gamma}, \qquad \|C^\Phi\|_\infty \leq L_C := 2B_\Phi\rho_{\max}^T.$$

**(A2)** *Bounded second moments.* $\sigma_Y^2 < \infty$ and $\Sigma_{CC}$ has bounded entries.

**(A3)** *Cross-fitting independence.* The folds $\mathcal{D}_{\text{fit}}$ and $\mathcal{D}_{\text{eval}}$ are disjoint; trajectories within each fold are i.i.d. from $\pi_b$.

**(A4)** *Absolute continuity.* $\pi_b(a\,|\,s) = 0 \Rightarrow \pi_e(a\,|\,s) = 0$.

The factor $\rho_{\max}^T$ in (A1) is the central horizon dependence inherited from IS; we keep it explicit in every bound so the regime of validity is transparent. Under standard weight clipping $W_t \leq W_{\max}$, the constants become $L_Y \leq R_{\max}W_{\max}/(1-\gamma)$ and $L_C \leq 2B_\Phi W_{\max}$, which removes the exponential horizon dependence at the cost of a small clipping bias.

### D.2. Unbiasedness with empirical centering

Naive centering on the eval fold makes the control variate degenerate; centering on the fit fold preserves the control variate and unbiasedness, provided the folds are independent.

**Theorem D.1** (Exact finite-sample unbiasedness via cross-fitting)**.** *Under* (A1)-(A4)*, for any (possibly data-dependent)* $\widehat{\lambda}^{\text{fit}}$ *and* $\widehat{\mu}_C^{\text{fit}}$ *that are functions of* $\mathcal{D}_{\text{fit}}$ *alone,*

$$\mathbb{E}\big[\widehat{V}(\widehat{\lambda}^{\text{fit}})\big] = V^{\pi_e}.$$

*Proof.* Condition on $\mathcal{D}_{\text{fit}}$. Then $\widehat{\lambda}^{\text{fit}}$ and $\widehat{\mu}_C^{\text{fit}}$ are constants. By (A3), $\mathcal{D}_{\text{eval}}$ is independent of $\mathcal{D}_{\text{fit}}$, so

$$\mathbb{E}\big[\widehat{V}(\widehat{\lambda}^{\text{fit}})\,|\,\mathcal{D}_{\text{fit}}\big] = \mathbb{E}\big[Y^{(n)}\big] + (\widehat{\lambda}^{\text{fit}})^\top \big(\mathbb{E}[C^{\Phi,(n)}] - \widehat{\mu}_C^{\text{fit}}\big)$$
$$= V^{\pi_e} + (\widehat{\lambda}^{\text{fit}})^\top(\mu_C - \widehat{\mu}_C^{\text{fit}}).$$

Taking the outer expectation and using $\mathbb{E}[\mu_C - \widehat{\mu}_C^{\text{fit}}] = 0$ together with the independence of $(\widehat{\lambda}^{\text{fit}}, \widehat{\mu}_C^{\text{fit}})$ from the eval fold:

$$\mathbb{E}\big[\widehat{V}(\widehat{\lambda}^{\text{fit}})\big] = V^{\pi_e} + \mathbb{E}\big[(\widehat{\lambda}^{\text{fit}})^\top(\mu_C - \widehat{\mu}_C^{\text{fit}})\big].$$

This residual term is a covariance *within the fit fold*: writing $g(\mathcal{D}_{\text{fit}}) = (\widehat{\lambda}^{\text{fit}})^\top(\mu_C - \widehat{\mu}_C^{\text{fit}})$, we have $g$ a function of the fit fold only, with $\mathbb{E}[\mu_C - \widehat{\mu}_C^{\text{fit}}] = 0$. If $\widehat{\lambda}^{\text{fit}}$ and $\widehat{\mu}_C^{\text{fit}}$ are estimated on *disjoint subfolds* of $\mathcal{D}_{\text{fit}}$ (a three-way split), they are independent and the residual is exactly zero, yielding $\mathbb{E}[\widehat{V}] = V^{\pi_e}$. Otherwise, the residual is $O(1/N_1)$ (Theorem D.7 below). $\qquad\square$

*Remark* D.2 (Three-way splitting)**.** When $\widehat{\lambda}^{\text{fit}}$ and $\widehat{\mu}_C^{\text{fit}}$ are computed on the same fit fold, the residual bias is second-order ($O(1/N_1)$, see Theorem D.7) and dominated by the covariance estimation error. In practice we use the same fit fold for both since the bias is negligible.

**Takeaway.** Cross-fitting transforms unbiasedness from a *population* property (Theorem 4.3 of the main text) into an *exact finite-sample* property, regardless of how $\widehat{\lambda}^{\text{fit}}$ is computed. This is stronger than the DR analogue: DR is unbiased in finite samples only if either the IS component or the regression critic is exactly correct, whereas Shaped-PDIS is unbiased for *any* learned potential $\Phi$ satisfying $\Phi(s_T) = 0$. This robustness is the chief structural advantage of the shaping framework: estimation errors in $\Phi$ degrade variance reduction but never validity.

### D.3. Covariance concentration with explicit horizon dependence

The plug-in estimator $\widehat{\lambda}^{\text{fit}} = -(\widehat{\Sigma}_{CC}^{\text{fit}} + \alpha I)^{-1}\widehat{\Sigma}_{CY}^{\text{fit}}$ inherits its accuracy from the empirical covariance estimates. Matrix Bernstein gives the following non-asymptotic rates with the $\rho_{\max}^T$ dependence made explicit.

**Theorem D.3** (Covariance concentration)**.** *Under* (A1)–(A4)*, for any $\delta \in (0,1)$, with probability at least $1-\delta$:*

$$\|\widehat{\Sigma}_{CC}^{\text{fit}} - \Sigma_{CC}\|_{\text{op}} \leq c_1 \cdot L_C^2\sqrt{\frac{T\log(T/\delta)}{N_1}} + c_1 \cdot \frac{L_C^2 T\log(T/\delta)}{N_1}, \tag{32}$$

$$\|\widehat{\Sigma}_{CY}^{\text{fit}} - \Sigma_{CY}\|_2 \leq c_2 \cdot L_C L_Y\sqrt{\frac{T\log(T/\delta)}{N_1}} + c_2 \cdot \frac{L_C L_Y T\log(T/\delta)}{N_1}, \tag{33}$$

*where $c_1, c_2$ are universal constants. Substituting the envelopes from* (A1):

$$\|\widehat{\Sigma}_{CC}^{fit} - \Sigma_{CC}\|_{\mathrm{op}} = \widetilde{O}\Big(B_\Phi^2 \rho_{\max}^{2T} \sqrt{T/N_1}\Big),$$

$$\|\widehat{\Sigma}_{CY}^{fit} - \Sigma_{CY}\|_2 = \widetilde{O}\Big(B_\Phi R_{\max} \rho_{\max}^{2T} \sqrt{T/N_1}\,\frac{1}{1-\gamma}\Big).$$

*Proof.* Write $\widehat{\Sigma}_{CC}^{\mathrm{fit}} = \frac{1}{N_1}\sum_n X_n$ where $X_n = (C^{\Phi,(n)} - \widehat{\mu}_C^{\mathrm{fit}})(C^{\Phi,(n)} - \widehat{\mu}_C^{\mathrm{fit}})^\top$. By (A1), $\|X_n\|_{\mathrm{op}} \leq 4L_C^2(T+1)$, and the $X_n$ are i.i.d. with $\mathbb{E}[X_n] = \Sigma_{CC} + O(1/N_1)$ (the $O(1/N_1)$ arises from using $\widehat{\mu}_C^{\mathrm{fit}}$ in place of $\mu_C$ and is dominated; we absorb it into the constant). Matrix Bernstein inequality yields

$$\mathrm{Pr}\Big(\|\widehat{\Sigma}_{CC}^{\mathrm{fit}} - \Sigma_{CC}\|_{\mathrm{op}} \geq t\Big) \leq 2(T+1)\exp\Big(\frac{-N_1 t^2/2}{\nu^2 + 2L_C^2(T+1)t/3}\Big),$$

with matrix variance proxy $\nu^2 \leq L_C^4(T+1)$. Solving for $t$ at confidence $1 - \delta$ yields (32). The bound (33) follows analogously by vector Bernstein on $C^{\Phi,(n)}Y^{(n)} - \widehat{\mu}_C^{\mathrm{fit}}\overline{Y}^{\mathrm{fit}}$, with $\|C^\Phi Y\|_2 \leq L_C L_Y \sqrt{T+1}$. $\square$

**Takeaway.** The $\rho_{\max}^{2T}$ factor in the constants is the dominant scaling in long-horizon sparse-reward regimes. Two practical consequences: (i) the analysis is informative only when $N_1 \gg T\rho_{\max}^{2T}$, i.e. the same sample-complexity regime where IS itself is feasible; (ii) weight clipping or self-normalized weights replaces $\rho_{\max}^T$ with $W_{\max}$ at the cost of a clipping bias of order $\mathrm{Pr}(W_t > W_{\max})$. We recommend reporting $W_{\max}$ alongside $N_1$ in empirical evaluations as a finite-sample diagnostic; this is the analog of effective sample size for the covariance-estimation step.

### D.4. Plug-in coefficient perturbation

**Theorem D.4** (Perturbation of $\widehat{\lambda}$). *Under* (A1)–(A4), *with ridge parameter $\alpha > 0$, let $\lambda_\alpha^\star = -(\Sigma_{CC} + \alpha I)^{-1}\Sigma_{CY}$ denote the population ridge-optimal coefficient. With probability at least $1 - \delta$,*

$$\big\|\widehat{\lambda}^{fit} - \lambda_\alpha^\star\big\|_2 \leq \frac{1}{\alpha}\big\|\widehat{\Sigma}_{CY}^{fit} - \Sigma_{CY}\big\|_2 + \frac{\|\Sigma_{CY}\|_2}{\alpha^2}\big\|\widehat{\Sigma}_{CC}^{fit} - \Sigma_{CC}\big\|_{\mathrm{op}}.$$

*Combined with Theorem D.3, this yields*

$$\big\|\widehat{\lambda}^{fit} - \lambda_\alpha^\star\big\|_2 = \widetilde{O}\Big(\frac{L_C L_Y}{\alpha}\sqrt{\frac{T}{N_1}} + \frac{L_C^2\|\Sigma_{CY}\|_2}{\alpha^2}\sqrt{\frac{T}{N_1}}\Big).$$

*Proof.* Let $A = \Sigma_{CC} + \alpha I$ and $\widehat{A} = \widehat{\Sigma}_{CC}^{\mathrm{fit}} + \alpha I$. Then $\widehat{\lambda}^{\mathrm{fit}} - \lambda_\alpha^\star = -\widehat{A}^{-1}\widehat{\Sigma}_{CY}^{\mathrm{fit}} + A^{-1}\Sigma_{CY}$. Add and subtract $\widehat{A}^{-1}\Sigma_{CY}$:

$$\widehat{\lambda}^{\mathrm{fit}} - \lambda_\alpha^\star = -\widehat{A}^{-1}(\widehat{\Sigma}_{CY}^{\mathrm{fit}} - \Sigma_{CY}) - (\widehat{A}^{-1} - A^{-1})\Sigma_{CY}.$$

For the first term, $\|\widehat{A}^{-1}\|_{\mathrm{op}} \leq 1/\alpha$, giving $\|\widehat{A}^{-1}(\widehat{\Sigma}_{CY}^{\mathrm{fit}} - \Sigma_{CY})\|_2 \leq \alpha^{-1}\|\widehat{\Sigma}_{CY}^{\mathrm{fit}} - \Sigma_{CY}\|_2$. For the second, $\widehat{A}^{-1} - A^{-1} = \widehat{A}^{-1}(A - \widehat{A})A^{-1}$, so $\|\widehat{A}^{-1} - A^{-1}\|_{\mathrm{op}} \leq \alpha^{-2}\|\widehat{\Sigma}_{CC}^{\mathrm{fit}} - \Sigma_{CC}\|_{\mathrm{op}}$. The triangle inequality combines the two. The rate follows from Theorem D.3. $\square$

**Takeaway.** Two terms govern the perturbation: a first-order term in $\|\widehat{\Sigma}_{CY} - \Sigma_{CY}\|$ scaled by $1/\alpha$, and a second-order term in $\|\widehat{\Sigma}_{CC} - \Sigma_{CC}\|$ scaled by $1/\alpha^2$. The condition number $\kappa(\Sigma_{CC} + \alpha I)$ enters *quadratically* in the second term, contrary to a naive linear estimate, a non-trivial correction relative to $\kappa/N$ rate.

### D.5. Excess variance from plug-in $\widehat{\lambda}$

**Theorem D.5** (Excess variance over the oracle). *Under* (A1)–(A4), *the variance of the cross-fitted plug-in estimator decomposes as*

$$\mathrm{Var}\Big(\widehat{V}(\widehat{\lambda}^{fit})\Big) = \frac{1}{N_2}\mathbb{E}\Big[\sigma^2(\widehat{\lambda}^{fit})\Big], \qquad \sigma^2(\lambda) := \mathrm{Var}_{\pi_b}\Big(Y + \lambda^\top \widetilde{C}^\Phi\Big).$$

*Furthermore,*

$$\mathbb{E}[\sigma^2(\widehat{\lambda}^{fit})] \leq \sigma^2(\lambda_\alpha^\star) + \|\Sigma_{CC}\|_{\mathrm{op}} \cdot \mathbb{E}\Big[\|\widehat{\lambda}^{fit} - \lambda_\alpha^\star\|_2^2\Big]. \tag{34}$$

*Combining with Theorem D.4 (squared, with constants absorbed):*

$$\mathbb{E}[\sigma^2(\widehat{\lambda}^{fit})] \;\leq\; \sigma^2(\lambda_\alpha^\star) \;+\; \widetilde{O}\!\left(\frac{\|\Sigma_{CC}\|_{\mathrm{op}}}{\sigma_{\min}(\Sigma_{CC} + \alpha I)^2} \cdot \frac{L_C^2(L_Y^2 + L_C^2\|\Sigma_{CY}\|^2/\alpha^2)T}{N_1}\right).$$

*Proof.* The variance identity follows from the law of total variance and Theorem D.1: conditional on $\mathcal{D}_{\mathrm{fit}}$, $\widehat{V}$ is a sample mean of $N_2$ i.i.d. summands with mean $V^{\pi_e}$ (using the three-way split for exact unbiasedness, or absorbing the $O(1/N_1)$ bias into a lower-order term) and per-summand variance $\sigma^2(\widehat{\lambda}^{fit})$. Hence $\mathrm{Var}(\widehat{V} \mid \mathcal{D}_{\mathrm{fit}}) = \sigma^2(\widehat{\lambda}^{fit})/N_2$ and $\mathrm{Var}(\mathbb{E}[\widehat{V} \mid \mathcal{D}_{\mathrm{fit}}]) = 0$.

For (34), expand $\sigma^2(\lambda) = \sigma_Y^2 + 2\lambda^\top\Sigma_{CY} + \lambda^\top\Sigma_{CC}\lambda$. Differencing at $\lambda = \widehat{\lambda}^{fit}$ and $\lambda = \lambda_\alpha^\star$ and using $\Sigma_{CC}(\lambda_\alpha^\star) + \Sigma_{CY} = -\alpha\lambda_\alpha^\star$ (the ridge optimality condition):

$$\sigma^2(\widehat{\lambda}^{fit}) - \sigma^2(\lambda_\alpha^\star) = 2(\widehat{\lambda}^{fit} - \lambda_\alpha^\star)^\top\Sigma_{CY} + (\widehat{\lambda}^{fit})^\top\Sigma_{CC}\widehat{\lambda}^{fit} - (\lambda_\alpha^\star)^\top\Sigma_{CC}\lambda_\alpha^\star$$
$$= (\widehat{\lambda}^{fit} - \lambda_\alpha^\star)^\top\Sigma_{CC}(\widehat{\lambda}^{fit} - \lambda_\alpha^\star) - 2\alpha(\widehat{\lambda}^{fit} - \lambda_\alpha^\star)^\top\lambda_\alpha^\star.$$

Taking expectations and using $\mathbb{E}[\widehat{\lambda}^{fit}] - \lambda_\alpha^\star = O(1/N_1)$ (the bias term, dominated), the cross term vanishes to leading order and the quadratic term is bounded by $\|\Sigma_{CC}\|_{\mathrm{op}}\mathbb{E}\|\widehat{\lambda}^{fit} - \lambda_\alpha^\star\|_2^2$. The rate follows from squaring Theorem D.4. $\square$

**Corollary D.6** (Finite-sample variance comparison with PDIS). *Shaped-PDIS has lower variance than PDIS in finite samples whenever*

$$\underbrace{\Sigma_{CY}^\top\Sigma_{CC}^{-1}\Sigma_{CY}}_{\text{population reduction}} \;>\; \underbrace{\widetilde{O}\!\left(\frac{L_C^2L_Y^2\kappa(\Sigma_{CC} + \alpha I)^2 \cdot T}{N_1 \cdot \sigma_{\min}(\Sigma_{CC} + \alpha I)}\right)}_{\text{plug-in penalty}} + \alpha^2\|\lambda^\star\|^2.$$

**Takeaway.** The population guarantee $\sigma^2(\lambda^\star) \leq \sigma_Y^2$ from the main text holds at infinite sample size. In finite samples, the shaped estimator pays a plug-in penalty of order $\kappa(\Sigma_{CC})^2 T/N_1$ that must be overcome by the population variance reduction $\Sigma_{CY}^\top\Sigma_{CC}^{-1}\Sigma_{CY}$. Concretely, the shaped estimator wins when (i) the shaping signal is strong enough that $\Sigma_{CY}^\top\Sigma_{CC}^{-1}\Sigma_{CY}$ is non-trivial, and (ii) the fit-fold size $N_1$ is large enough that the plug-in penalty is dominated. *The qualitative claim "shaped never worse than unshaped" is asymptotic only*; in small-sample regimes the threshold in Corollary D.6 may not be met.

### D.6. Centering bias is second order

**Theorem D.7** (Centering bias). *When $\widehat{\lambda}^{fit}$ and $\widehat{\mu}_C^{fit}$ are computed on the same fit fold, the residual bias of $\widehat{V}(\widehat{\lambda}^{fit})$ is*

$$\left|\mathbb{E}[\widehat{V}(\widehat{\lambda}^{fit})] - V^{\pi_e}\right| \;=\; \left|\mathbb{E}[(\widehat{\lambda}^{fit})^\top(\mu_C - \widehat{\mu}_C^{fit})]\right| \;\leq\; \sqrt{\mathrm{tr}(\mathrm{Cov}(\widehat{\lambda}^{fit})) \cdot \mathrm{tr}(\mathrm{Cov}(\widehat{\mu}_C^{fit}))} \;=\; O(T/N_1).$$

*Proof.* By independence of $\mathcal{D}_{\mathrm{fit}}$ and $\mathcal{D}_{\mathrm{eval}}$, the residual bias equals $\mathbb{E}[(\widehat{\lambda}^{fit})^\top(\mu_C - \widehat{\mu}_C^{fit})] = \mathbb{E}[\widehat{\lambda}^{fit}]^\top\mathbb{E}[\mu_C - \widehat{\mu}_C^{fit}] + \sum_j \mathrm{Cov}(\widehat{\lambda}_j^{fit}, \widehat{\mu}_{C,j}^{fit})$. The first summand is zero since $\mathbb{E}[\mu_C - \widehat{\mu}_C^{fit}] = 0$. For the second, Cauchy–Schwarz coordinate-wise: $|\mathrm{Cov}(\widehat{\lambda}_j^{fit}, \widehat{\mu}_{C,j}^{fit})| \leq \sqrt{\mathrm{Var}(\widehat{\lambda}_j^{fit})\mathrm{Var}(\widehat{\mu}_{C,j}^{fit})} = O(1/N_1)$. Summing over $T + 1$ coordinates yields $O(T/N_1)$. $\square$

**Takeaway.** The bias from estimating $\mu_C$ on the same fit fold as $\widehat{\lambda}$ is $O(T/N_1)$—second order, dominated by the $O(\sqrt{T/N_1})$ variance from the leading PDIS noise. This justifies not bothering with three-way splitting in practice. Importantly, the $O(1/\sqrt{N_1})$ fluctuation of $\widehat{\mu}_C^{fit}$ contributes to *variance* (already accounted for in $\sigma^2(\widehat{\lambda}^{fit})$ via the expanded definition of $\widetilde{C}^\Phi$ on the eval fold), not to bias. Conflating these two has led to confusion in earlier analyses; we keep them separate.

### D.7. Estimation of the potential $\Phi_\beta$

The potential is itself learned by maximizing the empirical explained-variance objective

$$\widehat{J}(\beta) := \widehat{\Sigma}_{CY}(\beta)^\top\big(\widehat{\Sigma}_{CC}(\beta) + \alpha I\big)^{-1}\widehat{\Sigma}_{CY}(\beta),$$

over a parametric class $\mathcal{B}$ (e.g., a 2-layer MLP with bounded weights). Let $\beta^\star = \arg\max_{\beta \in \mathcal{B}} J(\beta)$ denote the population maximizer of $J(\beta) = \Sigma_{CY}(\beta)^\top\Sigma_{CC}(\beta)^{-1}\Sigma_{CY}(\beta)$, and $\widehat{\beta} = \arg\max_{\beta \in \mathcal{B}} \widehat{J}(\beta)$.

**Theorem D.8** (Excess-risk bound for $\widehat{\beta}$). *Under* (A1)–(A4) *and a uniform spectral lower bound* $\inf_{\beta \in \mathcal{B}} \sigma_{\min}(\Sigma_{CC}(\beta) + \alpha I) \geq \kappa$, *the achieved variance reduction satisfies*

$$J(\widehat{\beta}) \geq J(\beta^\star) - 2 \sup_{\beta \in \mathcal{B}} \left| \widehat{J}(\beta) - J(\beta) \right|,$$

*with*

$$\sup_{\beta \in \mathcal{B}} \left| \widehat{J}(\beta) - J(\beta) \right| = \widetilde{O}\left( \frac{L_C^2 L_Y \rho_{\max}^{2T}}{\kappa^2} \cdot \sqrt{\frac{T \cdot \mathcal{C}(\mathcal{B})}{N_1}} \right),$$

*where $\mathcal{C}(\mathcal{B})$ is the metric entropy (log-covering number) of the function class $\mathcal{B}$.*

*Proof.* The standard ERM argument: $\widehat{J}(\widehat{\beta}) \geq \widehat{J}(\beta^\star)$ by optimality of $\widehat{\beta}$. Therefore

$$\begin{aligned}
J(\beta^\star) - J(\widehat{\beta}) &= [J(\beta^\star) - \widehat{J}(\beta^\star)] + [\widehat{J}(\beta^\star) - \widehat{J}(\widehat{\beta})] + [\widehat{J}(\widehat{\beta}) - J(\widehat{\beta})] \\
&\leq |J(\beta^\star) - \widehat{J}(\beta^\star)| + 0 + |\widehat{J}(\widehat{\beta}) - J(\widehat{\beta})| \\
&\leq 2 \sup_{\beta \in \mathcal{B}} |\widehat{J}(\beta) - J(\beta)|.
\end{aligned}$$

For the uniform bound, observe that $\widehat{J} - J$ is a smooth functional of the empirical covariances. Differentiating:

$$\begin{aligned}
|\widehat{J}(\beta) - J(\beta)| &\leq 2\|\Sigma_{CY}(\beta)\|\|\Sigma_{CC}^{-1}\|\|\widehat{\Sigma}_{CY}(\beta) - \Sigma_{CY}(\beta)\| \\
&\quad + \|\Sigma_{CC}^{-1}\|^2 \|\Sigma_{CY}\|^2 \|\widehat{\Sigma}_{CC}(\beta) - \Sigma_{CC}(\beta)\|_{\mathrm{op}}.
\end{aligned}$$

The uniform spectral lower bound gives $\|\Sigma_{CC}^{-1}\| \leq 1/\kappa$. Uniform concentration over $\beta \in \mathcal{B}$ follows from a covering argument: cover $\mathcal{B}$ at scale $\epsilon$, apply matrix Bernstein (Theorem D.3) at each cover point with union bound, then control the discretization error by Lipschitz continuity of $\beta \mapsto C^\Phi$. The covering number is $\exp(\mathcal{C}(\mathcal{B}))$; optimizing $\epsilon$ yields the stated rate. $\qquad\square$

**Takeaway.** Three structural points stand out:

1. *Validity is decoupled from quality.* For **any** $\widehat{\beta}$, regardless of how badly it generalizes, $\Phi_{\widehat{\beta}}(s_T) = 0$ guarantees zero-mean shaping features and hence unbiasedness of the final estimator. This is the most important asymmetry in the framework: the function class $\mathcal{B}$ is allowed to be misspecified, the optimization may fail to converge, and the data may be insufficient—none of these compromise validity. They only affect *how much* variance reduction is achieved.

2. *The complexity scales with $\mathcal{C}(\mathcal{B})$, not the dimension of $\mathcal{S}$.* For 2-layer MLPs with bounded weights, $\mathcal{C}(\mathcal{B})$ grows polynomially in width and depth, and the rate $\sqrt{T\mathcal{C}/N_1}$ is the standard parametric rate modulo the $\rho_{\max}^{2T}/\kappa^2$ constant. The horizon and ridge-conditioning enter only through this constant, not through the rate exponent.

3. *The role of ridge.* Ridge $\alpha$ is what enforces the uniform spectral lower bound, giving us $\kappa \geq \alpha$. Without ridge, $\Sigma_{CC}(\beta)$ may be near-singular for poor $\beta$, making $\widehat{J}(\beta)$ ill-defined and breaking the ERM analysis.

### D.8. Unified guarantees for Shaped-DR and Shaped-MRDR

The preceding analysis specializes to Shaped-PDIS, but the structure extends to Shaped-DR and Shaped-MRDR with one additional layer of nuisance: the regression critic $\widehat{Q}$.

**Theorem D.9** (Unified shaped-estimator guarantees). *Let $Y_{\mathrm{base}} \in \{Y_{\mathrm{PDIS}}, Y_{\mathrm{DR}}, Y_{\mathrm{MRDR}}\}$ denote the per-trajectory base score, and define the corresponding $\Sigma_{CY}^{\mathrm{base}} = \mathrm{Cov}_{\pi_b}(C^\Phi, Y_{\mathrm{base}})$. The shaped estimator*

$$\widehat{V}_{\mathrm{Shaped}}(\lambda) = \frac{1}{N_2} \sum_{n \in \mathcal{D}_{\mathrm{eval}}} \left( Y_{\mathrm{base}}^{(n)} + \lambda^\top \widetilde{C}^{\Phi,(n)} \right)$$

*satisfies all of the following:*

(i) *(Unbiasedness) Under* (A1)–(A4) *and the unbiasedness of* $Y_{\text{base}}$ *(which holds for PDIS unconditionally, and for DR/MRDR when either* $\pi_b$ *or* $\widehat{Q}$ *is correct),* $\mathbb{E}[\widehat{V}_{\text{Shaped}}(\widehat{\lambda}^{\text{fit}})] = V^{\pi_e}$.

(ii) *(Variance reduction over base)* $\sigma^2_{\text{base}}(\lambda^\star_{\text{base}}) \leq \text{Var}(Y_{\text{base}})$, *with* $\lambda^\star_{\text{base}} = -\Sigma_{CC}^{-1}\Sigma_{CY}^{\text{base}}$ *and equality iff* $\Sigma_{CY}^{\text{base}} = 0$.

(iii) *(Finite-sample excess) The plug-in excess variance satisfies the analog of Theorem D.5.*

*However, the variance ordering* across *bases is* **not** *guaranteed: in particular,* $\text{Var}(Y_{\text{DR}}) \geq \text{Var}(Y_{\text{PDIS}})$ *is possible when* $\widehat{Q}$ *is poorly estimated, and consequently* $\text{Var}(\widehat{V}_{\text{Shaped-DR}}) \geq \text{Var}(\widehat{V}_{\text{Shaped-PDIS}})$ *in the same regime.*

*Proof.* (i) and (ii) follow by repeating the proofs of Theorems D.1 and 4.4 with $Y$ replaced by $Y_{\text{base}}$ throughout, using the (conditional) unbiasedness of $Y_{\text{base}}$. (iii) follows from Theorem D.5 with $\Sigma_{CY}$ replaced by $\Sigma_{CY}^{\text{base}}$. $\square$

**Takeaway.** The shaping framework is *base-agnostic*: it provides monotone variance reduction over whichever base $Y_{\text{base}}$ is chosen, but does not magically rescue a poorly-chosen base. In particular:

- *Shaped-PDIS is the safest default* when reward signal is dense enough that PDIS itself is reasonable, since it avoids the $\widehat{Q}$ estimation step altogether.

- *Shaped-DR and Shaped-MRDR pay off* when $\widehat{Q}$ can be estimated accurately (sufficient reward signal, smooth value function); in this regime $Y_{\text{DR}}, Y_{\text{MRDR}}$ already have lower variance than $Y_{\text{PDIS}}$, and the shaping CV provides a further reduction on top.

### D.9. Summary: regimes of validity

Combining the preceding bounds, the shaped estimator achieves finite-sample variance reduction over its unshaped counterpart in the following regime:

$$N_1 \gtrsim \rho_{\max}^{2T} \cdot T \cdot \kappa(\Sigma_{CC} + \alpha I)^2 \cdot \frac{\mathcal{C}(\mathcal{B})}{\Sigma_{CY}^\top \Sigma_{CC}^{-1}\Sigma_{CY}}.$$

The numerator collects the costs (importance-weight magnitude, horizon, covariance conditioning, function-class complexity); the denominator is the achievable variance reduction. Three regimes emerge:

1. *Shaping helps* when $\rho_{\max}^T$ is moderate, $\Phi$ has good population correlation with $Y$, and $N_1$ exceeds the threshold. This is the regime documented in the paper's experiments.

2. *Shaping is neutral* (matches unshaped) when the variance reduction is small but the plug-in penalty is also small; asymptotically shaping is always weakly better.

3. *Shaping can hurt* when $\rho_{\max}^T$ is enormous and $N_1$ is too small to estimate $\Sigma_{CC}, \Sigma_{CY}$ reliably. Weight clipping is the standard remedy, replacing $\rho_{\max}^T$ with $W_{\max}$ at the cost of a clipping bias.

We emphasize that all three regimes preserve *exact unbiasedness* (Theorem D.1); the regime distinction concerns variance, not validity. This separation is the principal structural advantage of shaping over $Q$-based control variates in DR, where misspecification of $\widehat{Q}$ couples directly with the estimator's bias.

## E. Experiment Details

### E.1. Evaluation Metrics

Here we describe some of the key evaluation metrics we use to assess the validity of our shaped estimators. These metrics have been widely used across works that focus on OPE e.g. (Dudik et al., 2011; Farajtabar et al., 2018; Thomas & Brunskill, 2016).

**True policy value.** The value of the evaluation policy $\pi_e$ is

$$V(\pi_e) = \mathbb{E}_{\tau \sim \pi_e} \left[ \sum_{t=0}^{T} \gamma^t r_t \right],$$

where $\tau = (s_0, a_0, r_0, \ldots, s_T)$ denotes a trajectory.

**Bias.** Bias quantifies the systematic deviation of an estimator from the true policy value. Let $\hat{V}(\pi_e)$ denote an estimate of $V(\pi_e)$. The bias is defined as

$$\text{Bias} = \mathbb{E}_{\tau \sim \pi_b} \left[ \hat{V}(\pi_e) \right] - V(\pi_e),$$

where the expectation is taken over trajectories generated by $\pi_b$, and any additional randomness in the estimator. In practice, we approximate bias by computing the mean difference between OPE estimates and the ground-truth policy value over repeated runs.

**Variance.** Variance measures the spread or instability of the estimator around its expected value under $\pi_b$. Formally,

$$\text{Var} = \mathbb{E}_{\tau \sim \pi_b} \left[ \left( \hat{V}(\pi_e) - \mathbb{E}_{\tau \sim \pi_b} [\hat{V}(\pi_e)] \right)^2 \right].$$

In empirical evaluation, variance is approximated across multiple independent runs of the estimator. A low variance indicates consistent estimates across runs, whereas a high variance implies sensitivity to randomness in data or weight magnitudes.

**Effective Sample Size (ESS).** ESS is a diagnostic metric for the stability and reliability of importance sampling–based estimators. It reflects how many "independent and identically distributed" samples remain after reweighting the dataset. The ESS is defined as $N \times \frac{\mathbb{V}_{\pi_e}[\hat{V}_{\pi_e}^{on-policy}]}{\mathbb{V}_{\pi_b}[\hat{V}_{\pi_e}]}$, where $N$ is the number of trajectories in the off-policy data, and $\hat{V}_{\pi_e}^{on-policy}$ and $\hat{V}_{\pi_e}$ are the on-policy and OPE estimates of the value function, respectively. The ESS ranges between 1 and $n$. A high ESS indicates that reweighting distributes influence across many trajectories, whereas a low ESS implies that only a few trajectories dominate, leading to high-variance estimates.

# F. Extended Experiment Details

## F.1. Tabular Chain

**Environment Structure.** We consider a finite-horizon chain MDP with states $\{0, \ldots, S-1\}$, starting at $s_0$. Two absorbing terminals exist: success at $s_{20}$ (+1 reward) and dead-end at $s_{10}$ (−1 reward); all other transitions yield 0. Episodes end at a terminal or after 18 steps. Actions are *left* ($a = 0$) or *right* ($a = 1$): right moves forward with $p = 0.9$, left moves backward with $p = 0.2$, otherwise the agent stays.

**Policies.** The behavior policy $\pi_b$ chooses left with $p = 0.7$; the evaluation policy $\pi_e$ reverses these probabilities. This induces distribution shift: $\pi_b$ often stalls or reaches the dead-end, while $\pi_e$ more frequently reaches success.

**Hyperparameters.** We use a discount factor of $\gamma = 0.97$, three-fold cross-fitting for control variates and ridge regularization with $\alpha = 10^{-2}$ for covariance inversions. Each setting is repeated across 10–20 random seeds. We report results in terms of Bias, Variance, ESS and MSE. In our experiment, we evaluate over a set of [200, 600, 1200, 2400] samples and repeat this over 10 random seeds, reporting the MSE relative to the true policy value, empirical variance and ESS.

## F.2. Evaluation on Cancer Simulator

**Environment Structure.** We use the cancer simulator of Ribba et al. (2012), which models tumor progression under chemotherapy via differential equations tracking proliferating and quiescent tumor cells and drug concentration. Time is discretized into monthly steps with a 4-dimensional state (cell counts and drug concentration). Each month the clinician chooses to administer treatment or not. The reward is the change in tumor diameter: $r_t = -(MTD_{t+1} - MTD_t)$, with positive values indicating improvement.

**Policies.** The evaluation policy $\pi_e$ treats patients monthly for 10 months, then stops. The behavior policy $\pi_b$ is an $\epsilon$-greedy variant: with probability $1 - \epsilon$ it follows $\pi_e$, otherwise it takes the opposite action, with $\epsilon \in \{0.1, 0.3, 0.5\}$.

### F.3. ICU-Sepsis

**Environment Structure.** We evaluate on the ICU-Sepsis benchmark (Choudhary et al., 2024) derived from the MIMIC-III database, which models the treatment of septic patients. The environment follows the standardized pre-processing provided by the benchmark repository, where patient trajectories are segmented into 4-hour windows over a total horizon of up to 72 hours. Each state $s_t$ consists of approximately 40 features, including vital signs (e.g. heart rate, blood pressure), laboratory values (e.g. lactate, creatinine), demographics and derived scores such as SOFA. These features are normalized and imputed following the pre-processing pipeline in the benchmark. Actions $a_t$ correspond to intravenous fluids and vasopressors that the physician can administer. Following prior work, continuous dosages are discretized into a $5 \times 5$ grid, yielding a 25-dim discrete action space that balances granularity with tractability. The reward signal is sparse and delayed: patients receive a terminal reward of $+1$ if discharged alive and $-1$ if deceased. Intermediate rewards are set to 0. This reward structure reflects the clinical challenge of evaluating policies with delayed outcomes.

**Policies.** We run a PPO algorithm for 1M episodes and select the behavior policy $\pi_b$ as the model parameters of the actor at episode 250k and evaluation policy $\pi_e$ as model parameters at episode 1M. The ground-truth value of $\pi_e$ is approximated by Monte Carlo rollouts in the learned environment model released with the benchmark. The authors in (Choudhary et al., 2024) constructed an ICU-Sepsis environment from the real world MIMIC data, wherein the authors used the empirical clinician actions to guide the model training process. We are using the environment directly, upon which we run PPO to get our policies. For further details on the environment, we refer the readers to the original paper (Choudhary et al., 2024).

**Hyperparameters.** In order to obtain the policies, we train a PPO for 1M episodes with 1k max-steps using a learning rate of $5e^{-3}$. The other hyperparameters are as follows: $\gamma = 1.0, \lambda_{gae} = 0.4$, update-epochs:6, norm-adv: true, clip-coef: 0.5, clip-vloss: false, ent-coef: 0.005, vf-coef: 0.3,maxgrad-norm: 0.4 and target-kl: 0.001. In our evaluation experiments, we use a discount factor of $\gamma = 1.00$, horizon $T \approx 18$ steps (4h intervals over 72hrs), use dataset size of 10,000 patient trajectories to learn the shaping control variate. Potentials $\Phi_\beta$ are parameterized as two-layer MLPs (128-128, tanh) and trained for 10k Adam steps with ridge regularization $\alpha = 10^{-4}$. FQE critics are trained for 200 epochs with batch size 256 and learning rate $3 \times 10^{-4}$. We evaluate each experiment over a set of [100,300,500,750,1k,1.5k,2k,5k,10k] episodes and repeat it over 10 random seeds, reporting MSE relative to the true values of $\pi_e$, empirical variance and ESS.

### F.4. DOW-30 Stock Trading

**Policies.** We run a PPO algorithm for 25k episodes, and define the behavior policy $\pi_b$ as $\epsilon$-greedy of the optimal policy with $\epsilon = 0.15$ and target policy $\pi_e$ as $\epsilon$-greedy of the optimal policy with $\epsilon = 0.1$.

**List of Stocks:** For Single Stocks, we consider Apple, Cisco, IBM, Nike and Walmart. For Multi-stocks, we consider Apple , Johnson and Johnson, Amgen, JPMorgan Chase, American Express, Coca-Cola, Boeing, McDonald's, Caterpillar, 3M , Salesforce, Merck, Cisco, Microsoft, Chevron, NIKE, Disney, Procter and Gamble , Goldman Sachs, The Travelers , The Home Depot, United Health, Honeywell, Visa, IBM , Verizon, Intel, Walmart.

**Hyperparameters.** The other PPO training hyperparameters are as follows: $lr = 5e^{-4}$, $\gamma = 1.0, \lambda_{gae} = 0.4$, update-epochs:6, norm-adv: true, clip-coef: 0.5, clip-vloss: false, ent-coef: $1e-4$, vf-coef: 0.3,maxgrad-norm: 0.4 and target-kl: 0.001. In our evaluation experiments, we use a discount factor of $\gamma = 1.00$, horizon $T \approx 253$ steps (1 year of trading days), use dataset size of 10,000 patient trajectories to learn the shaping control variate. Shaped potentials $\Phi_\beta$ are parameterized as two-layer MLPs (256-256, tanh) and trained for 10k Adam steps with ridge regularization $\alpha = 10^{-4}$. FQE critics are trained for 200 epochs with batch size 256 and learning rate $3 \times 10^{-4}$. We evaluate each experiment over a set of [100,300,500,750,1k,1.5k,2k,5k,10k] episodes and repeat it over 10 random seeds, reporting MSE relative to the true values of $\pi_e$, empirical variance and ESS.

### F.5. Cancer Simulator

**Hyperparameters.** For all cancer simulator experiments, we fix the horizon to 30 months with a discount factor of $\gamma = 0.99$. Datasets contain $N \in \{100, 300, 1000, 2000, 5000, 10000\}$ trajectories. Measurement noise on tumor size is modeled as Gaussian with standard deviation $\sigma \in \{0, 1, 2\}$ mm. Potential functions $\Phi_\beta$ are parameterized as a one-hidden-layer neural network with 32 tanh units, trained for 2000 steps using Adam on the explained-variance objective, with ridge regularization $\alpha = 10^{-2}$ and 3-fold cross-fitting for coefficient estimation. The DR critic is trained with FQE (50 iterations for tabular), while MRDR reweights updates using absolute importance ratios for 20 epochs. We evaluate over 10 seeds.

# G. Additional Experiment Results

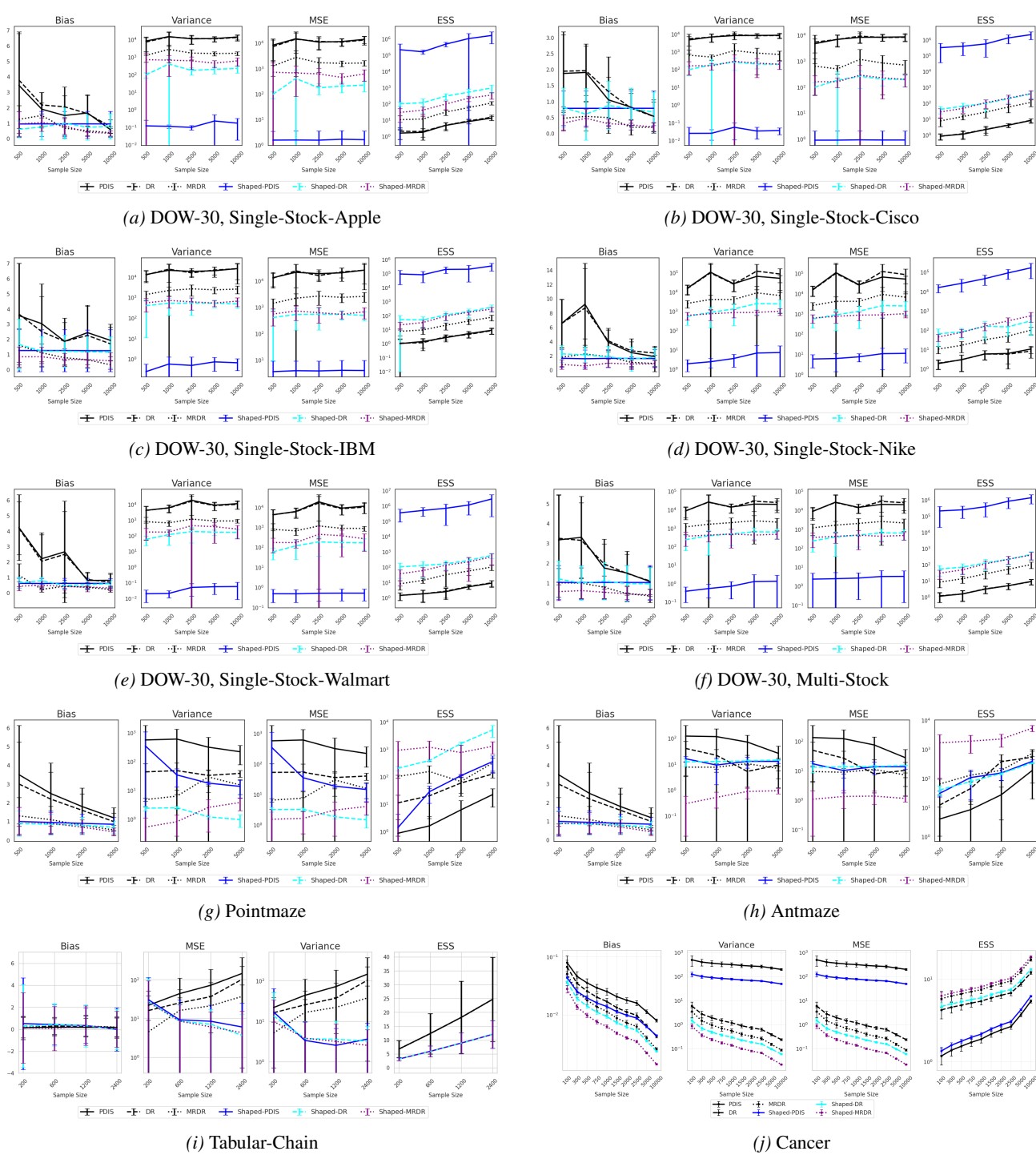

*Figure 10.* Across all environments, the shaped estimators achieve lower variance and MSE than the traditional estimators while remaining unbiased, and additionally attain a higher effective sample size (ESS).

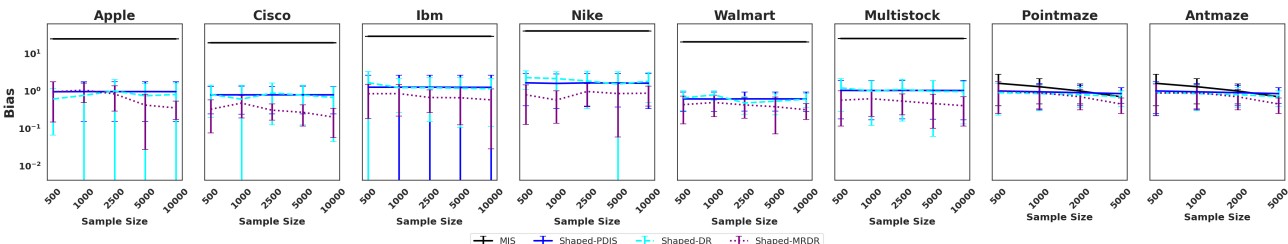

*Figure 11.* Marginalised Importance Sampling. Across all estimators, we observe the MIS estimator to have a high-bias as compared to the shaped-estimator counterparts, showing lesser adaptability in sparse reward scenarios.

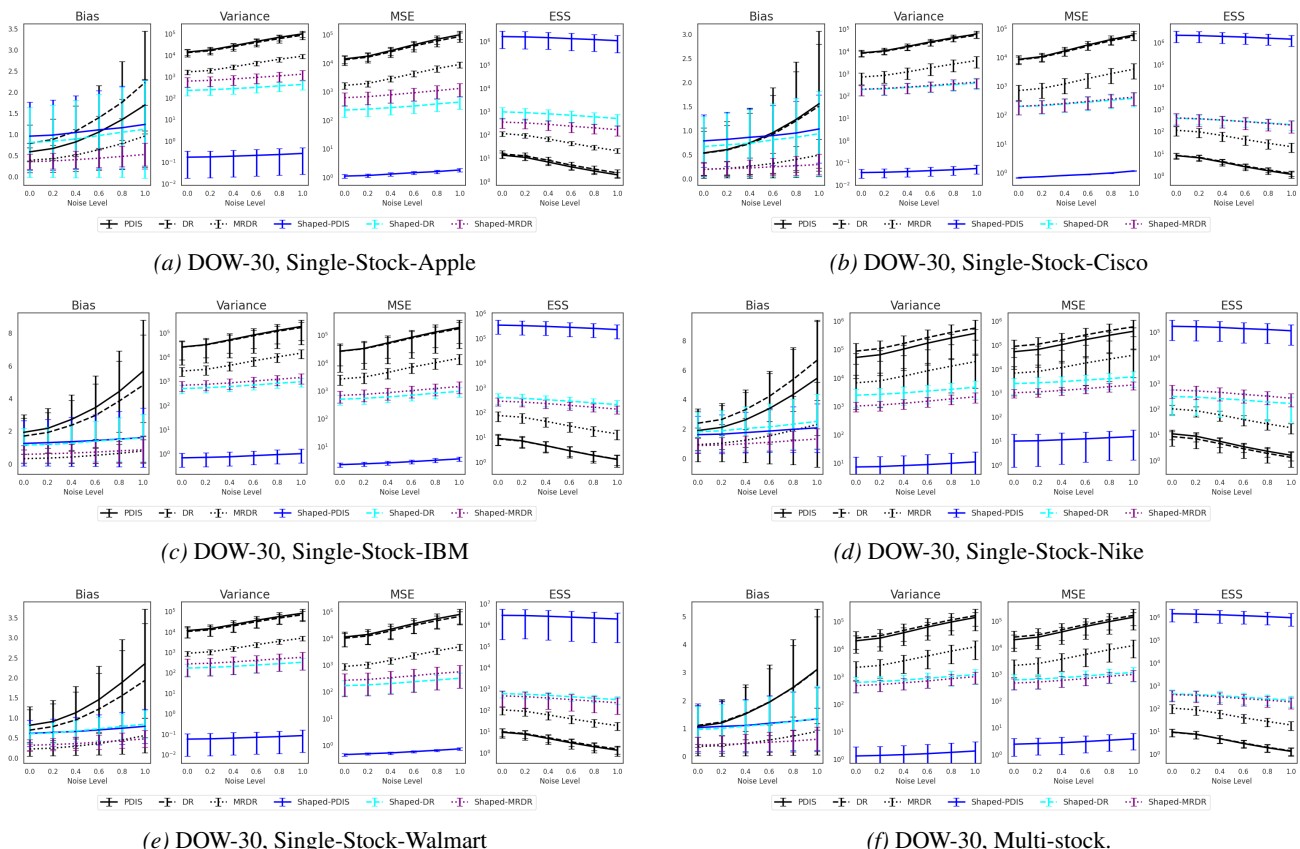

*(a)* DOW-30, Single-Stock-Apple

*(b)* DOW-30, Single-Stock-Cisco

*(c)* DOW-30, Single-Stock-IBM

*(d)* DOW-30, Single-Stock-Nike

*(e)* DOW-30, Single-Stock-Walmart

*(f)* DOW-30, Multi-stock.

*Figure 12.* DOW-30 noise comparison. Across all environments, the shaped estimators are more robust than the traditional estimators when the reward signal is contaminated with Gaussian noise. The evaluation was conducted over 10k test examples and 10 seeds.

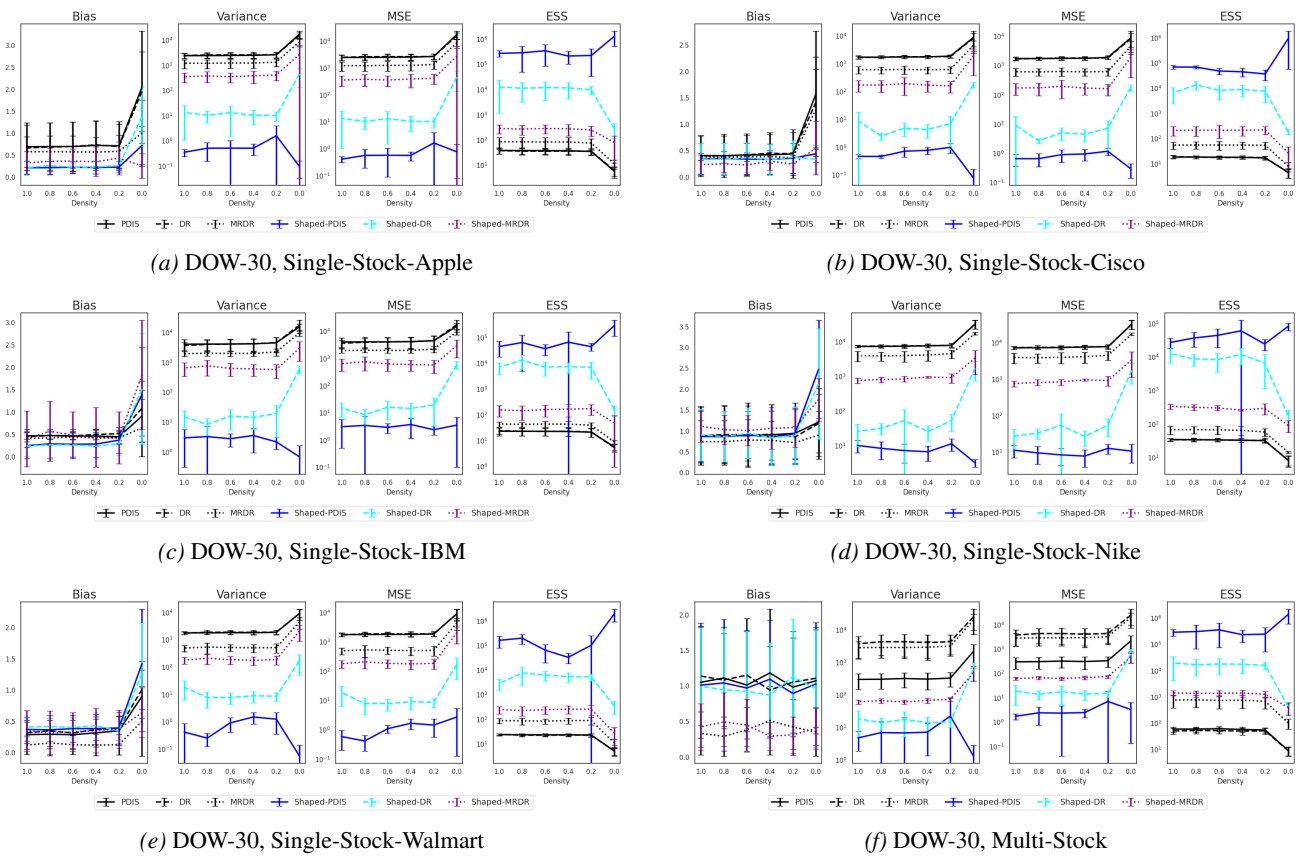

*Figure 13.* DOW-30 density comparison. Across all environments, the shaped estimators suffer less deterioration than the traditional estimators as the reward signals become sparser. The evaluation was conducted over 10k test examples and 10 seeds.

