# OpenReview forum: "Reward Shaping Control Variates for Off-Policy Evaluation Under Sparse Rewards"
_ICML.cc/2026/Conference — ICML 2026 regular_

### Official Review · Reviewer_W8Tq · 2026-02-28

**Soundness:** 2
**Presentation:** 2
**Significance:** 3
**Originality:** 3
**Overall Recommendation:** 3
**Confidence:** 3

**Summary:**

This paper introduces a new class of reward-shaping control variates for variance reduction in off-policy evaluation (OPE). Both theoretical guarantees and empirical studies are provided to show their improvements in domains under sparse rewards.

**Compliance With Llm Reviewing Policy:**

Affirmed.

**Final Justification:**

I appreciate the rebuttal and the additional clarifications. The claim of “orders of improvement” in variance reduction is not fully convincing to me, as the current analysis characterizes the proportion of improvement rather than establishing an order-level gain. Therefore, I maintain my original assessment.

**Key Questions For Authors:**

1. In Line 284 (right column), regarding the concentration bounds: what is the source of randomness in these bounds? Are they high-probability bounds over sampling randomness, or bounds in expectation?
2. Section 6 claims that shaped estimators have orders of improvement, while the theory only guarantees that the variance can be weakly reduced. Could the authors reconcile these empirical findings with their theoretical results?
3. Does the variance reduction property hold universally, even when rewards are not sparse? What is the precise relationship between sparsity and the magnitude of improvement?

**Limitations:**

yes

**Strengths And Weaknesses:**

Strengths
1. The potential-based reward shaping only needs correlation with the base estimator's return, which is a simpler target for learning.
2. The approach is interpretable and yields a lightweight variance-reduction technique for OPE.

Weaknesses
1. The paper lacks theoretical guarantees regarding asymptotic behavior with respect to the number of trajectories and their horizon length.
2. Section 6 claims “orders of improvement” in empirical studies, while the theoretical results only establish weak variance reduction. The empirical magnitude of improvement does not fully reconcile with the theoretical guarantees.
3. The sparsity of rewards is not formally characterized. The mechanism by which sparse rewards affect estimation difficulty, and how reward shaping specifically mitigates this issue, lacks clarification.

---

> ### Author Rebuttal · Authors · 2026-03-30
>
> We thank Reviewer W8Tq for their questions. We address them below in a sequential manner.
>
> Q1: The bounds are high-probability bounds over the sampling randomness of the $N$ i.i.d. trajectories from $\pi\_b$.
>
> Q2: We show in Theorem 4.4 and Corollary 4.5 that $$\text{Var}(\hat{V}\_{\text{Shaped}}(\lambda^\*)) \leq \text{Var}(\hat{V}\_{\text{base}}),$$ with strict inequality if and only if $\Sigma\_{CY} \neq 0$. This is deliberately a worst-case guarantee: it says the method can never hurt, regardless of the policies or the potential function. The theorem does not place an upper bound on how much variance reduction is achievable, it only provides a lower bound (zero reduction) on the improvement. The actual variance reduction achieved is given by the explained-variance term:
> $$\Delta\text{Var} = (\Sigma\_{CY}^{\text{base}})^\top \Sigma\_{CC}^{-1} \Sigma\_{CY}^{\text{base}}.$$
> This quantity can range from negligibly small (when shaping features are nearly uncorrelated with the return) to nearly equal to $\text{Var}(Y\_{\text{base}})$ (when shaping features capture almost all of the return's variability). The theory guarantees $\Delta\text{Var} \geq 0$; the experiments show that in sparse-reward settings, $\Delta\text{Var}$ is in fact very large relative to $\text{Var}(Y\_{\text{base}})$.
> **Why the correlation $\Sigma\_{CY}$ is large under sparsity.** In a sparse-reward environment:
> - **Successful trajectories** (those that eventually receive $r\_{T-1} > 0$) tend to visit states with increasing $\Phi$ values, so $\gamma\Phi(s\_{t+1}) - \Phi(s\_t) > 0$ on average. These trajectories also have large positive $Y$.
> - **Failed trajectories** (those with $r\_t = 0$ everywhere) tend to visit states with decreasing or flat $\Phi$ values, so $\gamma\Phi(s\_{t+1}) - \Phi(s\_t) \leq 0$ on average. These trajectories have $Y = 0$.
> This creates a strong correlation between the shaping features $C^{\Phi}\_t = \gamma^t W\_t(\gamma\Phi(s\_{t+1}) - \Phi(s\_t))$ and the return $Y$: both are large for successful trajectories, and both are near zero for failed ones. The cross-covariance $\Sigma\_{CY}$ is therefore large in magnitude, and $\Delta\text{Var} = \Sigma\_{CY}^\top \Sigma\_{CC}^{-1} \Sigma\_{CY}$ captures a large fraction of $\text{Var}(Y)$. By contrast, for dense rewards, $Y$ is a sum of many nonzero terms spread across timesteps, and its variability is less concentrated in the success/failure distinction. In this regime, shaping features are still correlated with $Y$, but the correlation is weaker relative to the total variance, so the improvement, while still non-negative, is more modest.
>
> Q3: Yes, the variance reduction guarantee (Corollary 4.5) holds for any reward structure: $\text{Var}(\hat{V}\_{\text{Shaped}}(\lambda^\*)) \leq \text{Var}(\hat{V}\_{\text{base}})$, with strict inequality if $\Sigma\_{CY}^{\text{base}} \neq 0$. This requires only: (i) terminal boundary condition $\Phi(s\_T) = 0$, (ii) known $\pi\_b$ (Assumption 2), and (iii) $\Sigma\_{CC} \succ 0$, none requiring reward sparsity. Our density sweeps (Figures 7–8, 20–24) provide direct empirical evidence: as density $d$ decreases, traditional estimators degrade sharply due to fewer reward-carrying trajectories, while shaped estimators degrade gracefully because $\Phi$ extracts signal from state visitations across all trajectories. The gap between shaped and traditional estimators widens monotonically as $d$ decreases, confirming that sparser rewards yield larger relative improvements.
>
> W1: Please refer to our responses to Reviewers 7ZQd, RDwM for finite-sample proof sketches on covariance matrices and estimator mean. Here, we provide proof sketches here for \phi_\beta estimation guarantees in finite-sample scenarios. We can add a detailed proof in the camera-ready version.
>
> **$\Phi\_\beta$ estimation: proof sketch.** The potential $\Phi\_\beta$ is trained by maximizing the sample explained-variance objective $\hat{J}(\beta) = \hat{\Sigma}\_{CY}(\beta)^\top \hat{\Sigma}\_{CC}(\beta)^{-1} \hat{\Sigma}\_{CY}(\beta)$. Let $\Phi^\*$ denote the oracle-optimal potential maximizing the population counterpart $J^\*(\beta) = \Sigma\_{CY}(\beta)^\top \Sigma\_{CC}(\beta)^{-1} \Sigma\_{CY}(\beta)$. The achieved variance reduction at $\hat{\beta}$ satisfies:
>
> $$J^\*(\hat{\beta}) \geq \hat{J}(\hat{\beta}) - |J^\*(\hat{\beta}) - \hat{J}(\hat{\beta})|.$$
>
> The gap $|J^\*(\hat{\beta}) - \hat{J}(\hat{\beta})|$ is controlled by the uniform convergence of sample covariances to population covariances over the function class $\{\Phi\_\beta : \beta \in \mathcal{B}\}$. For parametric classes (e.g. neural networks), standard covering number or Rademacher complexity arguments yield $\sup\_{\beta \in \mathcal{B}} |J^\*(\beta) - \hat{J}(\beta)| = O\_p(\mathcal{C}(\mathcal{B})/\sqrt{N\_{\text{fit}}})$, where $\mathcal{C}(\mathcal{B})$ captures the complexity of the function class. Crucially, unbiasedness holds unconditionally for any $\Phi$ satisfying $\Phi(s\_T) = 0$ (Lemma 4.2).

---

> > ### Author Rebuttal · Reviewer_W8Tq · 2026-04-04
> >
> > Thank you for the rebuttal and the additional clarifications. While some points have been addressed, my concerns—particularly those in W2 and W3—remain unresolved. Therefore, I maintain my original assessment.

---

> > > ### Author Response · Authors · 2026-04-04
> > >
> > > We thank the reviewer for acknowleding our rebuttal and the opportunity to engage in the discussion. We would like to clarify further:
> > >
> > > 1. We respectfully note that **W2 and Q2 raise the identical concern**, reconciling the "orders of improvement" empirical results with the theoretical weak-inequality guarantee and we addressed this in our initial rebuttal. Since the reviewer indicates this remains unresolved, we expand upon our argument below.
> > >
> > > We show in Theorem 4.4 and Corollary 4.5 that $\text{Var}(\hat{V}\_{\text{Shaped}}(\lambda^\*)) \leq \text{Var}(\hat{V}\_{\text{base}})$, with strict inequality iff $\Sigma\_{CY} \neq 0$. This is deliberately a worst-case guarantee: the method can never hurt, regardless of the MDP, policies, or potential function. The theorem places no upper bound on variance reduction, only a lower bound (zero) on improvement. The actual reduction is:
> > >
> > > $$\Delta\text{Var} = (\Sigma_{CY}^{\text{base}})^\top \Sigma_{CC}^{-1} \Sigma_{CY}^{\text{base}},$$
> > >
> > > which ranges from negligibly small (shaping features uncorrelated with return) to nearly $\text{Var}(Y_{\text{base}})$ (shaping features capture almost all return variability). The theory guarantees $\Delta\text{Var} \geq 0$; experiments show that under sparsity, $\Delta\text{Var}$ is very large relative to $\text{Var}(Y_{\text{base}})$.
> > >
> > > **Why $\Sigma_{CY}$ is large under sparsity.** In sparse-reward environments: (1) *Successful trajectories* (receiving $r_{T-1} > 0$) visit states with increasing $\Phi$ values, so $\gamma\Phi(s_{t+1}) - \Phi(s_t) > 0$ on average, and have large positive $Y$. (2) *Failed trajectories* ($r_t = 0$ everywhere) visit states with flat/decreasing $\Phi$ values and have $Y = 0$. This co-movement makes $\Sigma_{CY}$ large in magnitude, and $\Delta\text{Var}$ captures a large fraction of $\text{Var}(Y)$. By contrast, under dense rewards, $Y$ is a sum of many nonzero terms spread across timesteps, the success/failure distinction is less sharp, and while improvement remains non-negative, it is more modest.
> > >
> > > **There is no gap to reconcile.** The theoretical guarantee is a *safety* result ($\Delta\text{Var} \geq 0$) holding universally, while the *magnitude* is problem-dependent and governed by $\Sigma_{CY}$. Sparse-reward structure creates exactly the conditions: A binary success/failure distinction that $\Phi$ captures, under which $\Sigma_{CY}$ is large and orders-of-magnitude reduction follows directly from the explained-variance formula.
> > >
> > > 2. W3: We acknowledge we couldn't answer this in our initial response due to character limits. We would like to address this here. We provide a formal characterization below:
> > >
> > > **Formal definition.** Define the trajectory reward support as $p\_{\text{supp}} := \mathbb{P}\_{\tau \sim \pi\_b}[\sum\_{t} |r\_t| > 0]$. All our domains have terminal-only rewards.
> > >
> > > **How sparsity degrades PDIS.** Under terminal-only rewards, $\text{Var}\_{\pi\_b}(Y) = p\_{\text{supp}} \cdot \gamma^{2(T-1)} \mathbb{E}[W\_{T-1}^2 \mid \text{success}] \cdot R^2 - (V^{\pi\_e})^2$, where $p\_{\text{supp}} = \mathbb{P}\_{\tau \sim \pi\_b}[\sum\_t |r\_t| > 0]$. Proof sketch for this: Expand Variance as E[X^2]- (E[X])^2 and rewrite in terms of rewards and shaped control variates. Only $\approx p\_{\text{supp}} \cdot N$ trajectories contribute nonzero $Y^{(n)}$, each with potentially enormous $W\_{T-1}$, creating a heavy-tailed distribution dominated by rare outliers.
> > >
> > > **How shaping mitigates this.** The shaping features $C^{\Phi}\_t = \gamma^t W\_t(\gamma\Phi(s\_{t+1}) - \Phi(s\_t))$ are nonzero at \*every\* timestep regardless of whether $r\_t = 0$, because they depend on states, not rewards. All $N$ trajectories contribute to shaping features, whereas only $\approx p\_{\text{supp}} \cdot N$ contribute to $Y$. This means $\Sigma\_{CC}$ is estimated from the full dataset, giving shaping a fundamental statistical advantage over DR's Q-function, which is effectively learned from only $p\_{\text{supp}} \cdot N$ informative trajectories. Under sparsity, successful trajectories produce both large positive $Y$ and large positive shaping increments, while failed trajectories produce $Y = 0$ and flat/negative increments, this makes $\Sigma\_{CY}$ large, yielding substantial $\Delta\text{Var} = \Sigma\_{CY}^\top \Sigma\_{CC}^{-1} \Sigma\_{CY}$.
> > >
> > > As an alternative view to density (in our original submission), we conduct density sweeps by removing the final terminal signal (inducing even more sparsity) in our density sweeps (Figures 7–8, 20–24). We show that Shaped estimators consistently outperform traditional counterparts and degrade more gracefully as sparsity increases, confirming their ability to extract signal from unsuccessful trajectories.
> > >
> > > We are happy to augment this analysis in the camera-ready version of the paper. Hopefully this addresses all your questions, and we are very happy to clarify any follow-up questions you might have.

---

### Official Review · Reviewer_qfjb · 2026-03-06

**Soundness:** 3
**Presentation:** 2
**Significance:** 3
**Originality:** 3
**Overall Recommendation:** 4
**Confidence:** 2

**Summary:**

This paper aims to address the challenge of the high variance in off-policy evaluation (OPE) under sparse reward settings. Existing approaches like Importance Sampling, Doubly Robust estimation, and Marginalized estimators often struggle with high variance under sparse reward or sensitivity to model inaccuracies. To tackle the issue under sparse reward, the method leverages potential-based reward shaping to generate zero-mean control variates. It ensures that the estimator remains unbiased regardless of the potential function's accuracy. The minimum variance of the estimator has an analytical solution, which serves as the loss function to train the neural network to learn the potential function. Empirical results demonstrate the lower variance and bias compared with existing baselines.

**Compliance With Llm Reviewing Policy:**

Affirmed.

**Final Justification:**

Authors answered my questions in the rebuttal. I keep my rating.

**Key Questions For Authors:**

N/A

**Limitations:**

To optimize the loss function for minimizing variance, the inverse of a matrix needs to be computed. This can be hard for high-dimensional data and sensitively relies on the regularization parameter $\alpha$.

**Strengths And Weaknesses:**

Strengths:

+ Soundness: The proposed method is in general sound. It is motivated by the potential based reward shaping, such that it can provide a dense reward signal for sparse reward setting. To ensure the unbiasedness, the shaped rewards are centered. To eliminate the variance, a loss function (with closed form) is derived, such that this control variate is optimized to have a high correlation with the original OPE target.

+ Presentation: The paper is in general easy to follow. Related works, i.e., Importance Sampling, Doubly Robust estimation, Marginalized estimators are discussed. The method is well motived. As stated, previous methods all suffer from high variance when faced with sparse reward, this paper tackles the problem by using potential based reward shaping, which does not requires to learn a Q function or to estimate state-visitation ratios as in previous methods.

+ Significance: The proposed method bridges potential-based reward shaping with control variate techniques, which provides a new insight for  minimizing variance in OPE. Also I feel this method is related to DR. DR methods usually require learning a Q function in advance. In contrast, the potential-based reward structure used in the proposed method also ensures potentially learning a value function.

Weakness:

+ Presentation: Based on my understanding, the proposed method is actually also learning a value function (tho through a variance minimization loss function). Since to serve as a control variate, $C^\Phi$ should have high correlation with $Y$, then when $\Phi(s)=V^{\pi_e}(s)$ the correlation is the highest. I saw the authors used a subsection (4.4) to discuss the relation between the proposed method and DR (which requires learning a Q function) but without mentioning the proposed method is potentially also learning a value function. Authors can consider adding more discussion if my point is correct.

Minor issues:
+ MRDR has no reference in the main paper, e.g., line 297. FQE (line 312 right column)has no explanation and no reference.
+ Line 252 right "See Appendix D for details". It is Appendix C I think.
+ I find the detailed discussion of Existing OPE estimators are put in Appendix A. It is better to move to the main text such that reader will be more clear of the literature.

(I am not an expert on OPE, so I might be unclear of some existing techniques)

---

> ### Author Rebuttal · Authors · 2026-03-30
>
> We thank the reviewer for their insights. We respond to the weaknesses below in a detailed manner:
>
> Response to Weakness:
>
> The intuition that $\phi$ may correlate with $V^{\pi_e}$ is partially correct, but the precise relationship is more nuanced, and the distinction is central to our method's advantages.
>
> **Why $\Phi \neq V^{\pi_e}$ in general.** The optimal $\Phi$* maximizes the explained-variance objective $J(\beta) = \Sigma_{CY}(\beta)^\top \Sigma_{CC}(\beta)^{-1} \Sigma_{CY}(\beta)$, which measures correlation between importance-weighted shaping differences and IS return fluctuations under $\pi_b$. This differs from $V^{\pi_e}$ in several fundamental ways: (1) $V^{\pi_e}$ minimizes Bellman error under $\pi_e$ dynamics, while $\Phi$* maximizes correlation with IS return fluctuations under $\pi_b$ trajectories; (2) the shaping features involve differences $\gamma\Phi(s_{t+1}) - \Phi(s_t)$, so $\Phi$ is defined only up to additive constants, its absolute value has no return interpretation; (3) $\Phi$* must account for the importance weight structure, favoring accurate shaping at early timesteps where $W_t$ is well-behaved over late timesteps where $W_t$ is noisy, a consideration absent from $V^{\pi_e}$.
>
> **What $\Phi$ actually learns.** Rather than a value function, $\Phi$ learns a scalar ranking of states distinguishing successful from unsuccessful trajectories as observed under $\pi_b$. In ICU-Sepsis (Tables 1–2), the learned $\Phi$ assigns high values to physiologically stable states and low values to deteriorating ones, which resembles a value function but is learned through a fundamentally different and easier pathway: every trajectory, including the majority that fail, provides dense training signal about state visitation patterns, whereas $V^{\pi_e}$ must extrapolate from rare non-zero rewards.
>
> **Why this distinction matters practically.** Errors in $\Phi$ cannot introduce bias. If $\Phi$ is poor, the optimal $\lambda$* simply reduces the shaping correction's weight (Corollary 4.5). By contrast, a poorly estimated $\hat{Q}$ in DR can increase variance with no self-correcting mechanism. So while $\Phi$ and $V^{\pi_e}$ may correlate in practice, conflating them obscures the method's core advantage: achieving a similar effect through a safer learning pathway.
>
> We will add a discussion after Section 4.4 in the revised manuscript clarifying this interpretation.
>
> Response to minor issues:
>
> Thank you for spotting the minor issues! We will fix the MRDR and Appendix references, and modify the related-work in the camera-ready version of the paper.
>
> Response to limitations:
>
> We never explicitly invert the covariance matrix. As described in Appendix C, we solve the regularized linear system $(\hat{\Sigma}\_{CC} + \alpha I)\lambda = -\hat{\Sigma}\_{CY}$ via the conjugate gradient (CG) method (Shewchuk et al., 1994), which requires only matrix-vector products and not the full inverse. The product is computed directly from the data as $\hat{\Sigma}\_{CC} v = \frac{1}{N} \sum\_n \tilde{C}^{\Phi,(n)} (\tilde{C}^{\Phi,(n)\top} v)$, costing $O(N(T+1))$ per iteration with no matrix materialization. Training $\Phi\_\Beta$ via backpropagation through $J(\beta)$ including ridge regression converged within approximately 500 iterations across all environments, including real-world ICU-Sepsis and non-stationary DOW-30 trading tasks.

---

> > ### Author Rebuttal · Reviewer_qfjb · 2026-04-01
> >
> > I thank authors for answering my questions.

---

> > > ### Author Response · Authors · 2026-04-04
> > >
> > > We sincerely thank the reviewer for acknowledging our rebuttal and help us significantly strengthen our paper, If you are satisfied with the rebuttals and the paper's contributions, both from a theoretical and a practical standpoint, would you consider raising the score?

---

### Official Review · Reviewer_RDwM · 2026-03-09

**Soundness:** 3
**Presentation:** 3
**Significance:** 2
**Originality:** 3
**Overall Recommendation:** 4
**Confidence:** 3

**Summary:**

This paper uses potential-based reward shaping to construct zero-mean control variates for OPE. The learned potential \phi(s) can reduce the variance for the importance sampling estimator and remain unbiased. Experiments on 9 environments show consistent variance/MSE improvements of the proposed estimators.

**Compliance With Llm Reviewing Policy:**

Affirmed.

**Final Justification:**

The authors have adequately addressed my main concerns. I will maintain my recommendation of weak accept.

**Key Questions For Authors:**

For Shaped-DR and Shaped-MRDR, the Q-function (via FQE) appears to be trained on the full dataset, while \phi and \lambda are cross-fitted on K folds? I'm wondering if sampling splitting is also needed for Q function to achieve the theoretical guarantee?

**Limitations:**

N/A.

**Strengths And Weaknesses:**

Strength:
1. The derivation of the proposed construction is easy to follow, and the motivation is well characterized.
2. The paper provides a clean population-level theory. It shows that PDIS is always unbiased for any \lambda, always better than IS and provides the closed-form solution for the optimal choice of \lambda.
3. The paper conducts broad experiments across many different environments. Especially, ICU-Sepsis interpretability analysis showing clinically meaningful \phi structure.

Weaknesses:
1.  All guarantees are oracle-level, missing finite-sample analysis. Especially, the covariance matrices estimation, the expectation estimation, and \phi_\beta estimation are not rigorously analyzed.  Also, one may need to consider the long-horizon issue, as it adds challenges in covariance estimations.
2. No marginalized estimator baselines in numerical experiments. The paper argues that DICE methods fail under sparsity, but never tests this. Marginalized estimators avoid trajectory IS's curse-of-horizon,  without comparison, it is unclear whether shaped-IS beats marginalized methods or just beats trajectory-IS.  Also, how is the proposed method compared to solely Q-learning or V-learning methods?
3. Experimental policy gaps are too small. DOW-30: \epsilon-greedy 0.15 vs 0.10 of the same optimal policy. Cancer: \epsilon-greedy perturbation of \pi_e. Near-on-policy settings with importance weights close to 1 don't reveal the difficulties of OPE in practice.

---

> ### Author Rebuttal · Authors · 2026-03-30
>
> We thank Reviewer RDwM for their questions. We address them below in a sequential manner.
>
> W1: We provide proof sketches below for covariance matrices and expectation estimation here, and \phi_\beta estimation as response to Reviewer W8Tq. We plan to include the full finite-sample analysis in the camera-ready version.
>
> **Proof sketch for finite-sample excess variance.** Let $\hat{\lambda} = -(\hat{\Sigma}\_{CC} + \alpha I)^{-1}\hat{\Sigma}\_{CY}$ denote the estimated coefficients. The actual variance decomposes as:
>
> $$\text{Var}(Y\_{\text{base}} + \hat{\lambda}^\top \tilde{C}^{\Phi}) = \underbrace{\text{Var}(Y\_{\text{base}} + \lambda^{*\top} \tilde{C}^{\Phi})}\_{\text{oracle variance}} + \underbrace{(\hat{\lambda} - \lambda^\*)^\top \Sigma\_{CC} (\hat{\lambda} - \lambda^\*)}\_{\text{excess from estimation}}.$$
>
> Under bounded moment assumptions on $\tilde{C}^{\Phi}$ and $Y\_{\text{base}}$, standard matrix concentration yields $\|\hat{\Sigma}\_{CC} - \Sigma\_{CC}\|\_{\text{op}}, \|\hat{\Sigma}\_{CY} - \Sigma\_{CY}\|\_2 = O\_p(\sqrt{(T+1)/N\_{\text{fit}}})$. A perturbation argument then gives $\|\hat{\lambda} - \lambda^\*\|\_2 = O\_p(\sqrt{(T+1)/N\_{\text{fit}}}/\sigma\_{\min}(\Sigma\_{CC}))$, so the excess variance is $O\_p(\kappa(\Sigma\_{CC})(T+1)/(N\_{\text{fit}} \cdot \sigma\_{\min}(\Sigma\_{CC})))$ and vanishes as $N\_{\text{fit}}/(T+1) \to \infty$. Ridge regularization ensures $\sigma\_{\min}(\hat{\Sigma}\_{CC} + \alpha I) \geq \alpha$, preventing this from exploding when $\Sigma\_{CC}$ is near-singular.
>
> **Mean centering: proof sketch.** The centering step subtracts the empirical mean $\hat{\mu}\_{C\_\beta} = \frac{1}{N}\sum\_{n=1}^N C^{\Phi\_\beta,(n)}$ from each shaping feature vector. By the central limit theorem, $\hat{\mu}\_{C\_\beta} - \mu\_{C\_\beta} = O\_p(1/\sqrt{N\_{\text{fit}}})$ componentwise. Cross-fitting ensures $\hat{\lambda}$ and $\hat{\mu}\_{C\_{\hat{\beta}}}$ are fixed constants on each evaluation fold (learned on separate data). The residual bias on each fold is:
>
> $$\mathbb{E}\_{\pi\_b}\left[Y^{(n)} + \hat{\lambda}^\top(C^{\Phi\_{\hat{\beta}},(n)} - \hat{\mu}\_{C\_{\hat{\beta}}})\right] = V^{\pi\_e} + \hat{\lambda}^\top(\mu\_{C\_{\hat{\beta}}} - \hat{\mu}\_{C\_{\hat{\beta}}}).$$
>
> This residual is $O\_p(\|\hat{\lambda}\|/\sqrt{N\_{\text{fit}}})$ per fold, and when averaged across $K$ folds further concentrates to $O\_p(\|\hat{\lambda}\|/\sqrt{N})$, vanishing asymptotically. The centering error propagates into $\hat{\Sigma}\_{CC}$ and $\hat{\Sigma}\_{CY}$ at the same $O\_p(1/\sqrt{N\_{\text{fit}}})$ rate, so it does not introduce an additional dominant term beyond the covariance estimation error.
>
> W2: In the ablation experiments over PointMaze and AntMaze asked by reviewer 7ZQD, we also compare the performance of MIS trained using DualDICE algorithm. We observe our shaped estimators not only beats trajectory IS but also Marginalised IS. As the author rebuttal duration was limited, we conducted MIS over these 2 examples, and are happy to add that baseline across all environments in the camera-ready version if all reviewers and ACs agree.
>
> W3: Thank you for the question. The trading environments are highly non-stationary, where even slight randomness leads to large changes in optimality and importance weights far from 1, compounded by long horizons ($T=253$ trading days). Below we show how on-policy values change across tasks with small changes in $\varepsilon$: $\varepsilon=0.1$ and $\varepsilon=0.15$ differ significantly in performance, and $\varepsilon=0.2$ approaches random performance. We will clarify this in the camera-ready version. For non-stationary environments (ICU-Sepsis, Pointmaze, Antmaze, and others), we use larger policy gaps and remain consistent.
>
> ### On-policy Values Across Stocks and Epsilon Values
>
> | $\varepsilon$ | Apple | Cisco | IBM | Nike | Walmart | Multi-stock |
> |---|-------|-------|-----|------|---------|-------------|
> | 0 | $3.88 \pm 0.02$ | $10.02 \pm 0.45$ | $6.01 \pm 0.14$ | $8.03 \pm 0.50$ | $5.90 \pm 0.56$ | $40.24 \pm 1.53$ |
> | 0.1 | $2.51 \pm 0.01$ | $6.64 \pm 0.27$ | $3.85 \pm 0.09$ | $5.13 \pm 0.36$ | $4.21 \pm 0.38$ | $24.71 \pm 1.17$ |
> | 0.15 | $1.70 \pm 0.01$ | $4.09 \pm 0.17$ | $2.29 \pm 0.05$ | $3.42 \pm 0.23$ | $2.34 \pm 0.19$ | $15.02 \pm 0.61$ |
> | 0.2 | $0.39 \pm 0.00$ | $0.83 \pm 0.03$ | $0.64 \pm 0.02$ | $0.72 \pm 0.04$ | $0.50 \pm 0.06$ | $1.83 \pm 0.28$ |
>
> Q1: For Shaped-DR and Shaped-MRDR, we train a separate FQE for each of the K folds alongside $\Phi$ and $\lambda$, ensuring all nuisance parameters are estimated on the fitting set and evaluated on the held-out fold.  This ensures consistency with the theoretical guarantees. This was not made explicit in the manuscript; we will add detailed algorithms for Shaped-DR and Shaped-MRDR in the Appendix.

---

> > ### Author Rebuttal · Reviewer_RDwM · 2026-04-03
> >
> > Thanks for the detailed review.

---

> > > ### Author Response · Authors · 2026-04-04
> > >
> > > We sincerely thank the reviewer for acknowledging our rebuttal and help us significantly strengthen our paper, If you are satisfied with the rebuttals and the paper's contributions, both from a theoretical and a practical standpoint, would you consider raising the score?

---

### Official Review · Reviewer_7ZQD · 2026-03-14

**Soundness:** 3
**Presentation:** 2
**Significance:** 2
**Originality:** 3
**Overall Recommendation:** 3
**Confidence:** 4

**Summary:**

This paper studies off-policy evaluation under sparse rewards. The authors propose a family of reward-shaping control variates based on potential-based shaping functions $\Phi(s)$, and use these control variates to augment standard OPE estimators such as PDIS, DR, and MRDR. The key idea is that although reward observations may be extremely sparse, state visitations are dense, so shaping terms of the form $\gamma \Phi(s_{t+1}) - \Phi(s_t)$ can provide a zero-mean correction that is correlated with the estimator noise and can therefore reduce variance without changing the target value. The paper proves unbiasedness of the shaped estimator, derives the variance-optimal coefficient vector $\lambda^\star$, and argues that combining shaping-based control variates with standard model-based control variates enlarges the variance-reduction subspace beyond DR and MRDR. Empirically, the method is evaluated on synthetic sparse-reward chains, a cancer simulator, several stock-trading environments, and an ICU-sepsis benchmark.

**Compliance With Llm Reviewing Policy:**

Affirmed.

**Ethical Review Concerns:**

I identified multiple bibliography entries that appear to contain fabricated or corrupted citation metadata, including at least one case where the cited arXiv identifier corresponds to an unrelated paper and another where the listed authorship does not match the cited work. I have already raised this in details in a private comment as well.

**Ethical Review Flag:**

Flag this paper for an ethics review.

**Ethics Expertise Needed:**

["Research Integrity Issues (e.g., plagiarism)"]

**Final Justification:**

I thank the authors for the detailed rebuttal and for providing the additional results. These experiments helpfully clarify the practical performance of the proposed reward-shaping control variates. However, my primary concerns regarding the lack of manual verification in the manuscript remain. Consequently, I maintain my original score.

**Key Questions For Authors:**

1. The empirical section would be significantly stronger if it included more canonical sparse-reward RL benchmarks such as PointMaze or AntMaze. Is there a reason these domains were not included? If the method performs well there too, that would materially strengthen my assessment of the paper’s significance and practical relevance.

2. The paper gives the cleanest variance guarantee for Shaped-PDIS. Could the authors state explicitly what population-level variance guarantees hold for Shaped-DR and Shaped-MRDR, especially once $\Phi$ and $\lambda$ are estimated from finite data? A more unified theorem statement or discussion would strengthen the paper.

3. How sensitive is performance to the parameterization and optimization of the potential function $\Phi_\beta$? Since the proposed method relies on learning $\Phi$ to correlate with return fluctuations, it would be helpful to know whether the empirical gains are robust across architectures and training settings, or whether performance depends strongly on careful tuning.

**Limitations:**

yes

**Strengths And Weaknesses:**

## Strengths

1. The paper addresses a central theme in off-policy evaluation: how to obtain reliable estimates in sparse-reward settings where standard IS- and DR-style estimators become difficult to use in practice. This is an important problem with clear relevance to applications such as healthcare and finance.

2. The core idea is interesting and reasonably original. Reward shaping and control variates are classical ideas individually, but their combination in this OPE form is meaningful, and the viewpoint of learning $\Phi$ for variance reduction rather than policy improvement is a useful contribution.

3. The paper provides a theoretical guarantee for the Shaped-PDIS estimator: with the optimal coefficient, the variance is guaranteed not to increase relative to the base estimator, with strict improvement when the control variate correlates with the return.

## Weaknesses

1. The paper’s framing around DR is sometimes imprecise. In the motivation, it can read as though DR becomes biased simply because the learned $Q$ function must extrapolate under sparse rewards, whereas in the appendix the paper itself follows the standard result that DR is unbiased when the behavior policy is known and has sufficient coverage for the target policy. I think the paper should distinguish more carefully between practical model misspecification issues and formal estimator bias.

2. The empirical evaluation misses canonical sparse-reward RL benchmarks such as *PointMaze or AntMaze*. For a paper whose central claim is strength in sparse long-horizon settings, this omission is noticeable and makes it harder to judge how broadly the method applies beyond the chosen synthetic and domain-specific domains.

3. While the learned potential $\Phi$ is central to the method, the paper does not fully characterize how sensitive performance is to the parameterization and optimization of $\Phi_\beta$. This matters because the practical success of the method may depend substantially on how well this auxiliary model is learned.

---

> ### Author Rebuttal · Authors · 2026-03-30
>
> We thank the Reviewer for the questions and address them below in a sequential manner.
>
> W2, Q1: We conduct our analysis on both PointMaze and AntMaze, and observe Shaped estimators to outperform vanilla estimators (including MIS using DualDICE). We used evaluation and behavior as 15% and 40% epsilon greedy over optimal policy, over 10 seeds.
>
> ### PointMaze MSE
>
> | Method | N=0.5k | N=1k | N=2k | N=5k |
> |--------|-------|--------|--------|--------|
> | PDIS | $579.55 \pm 1262.12$ | $616.25 \pm 761.66$ | $321.94 \pm 391.87$ | $223.30 \pm 144.58$ |
> | DR | $53.11 \pm 49.85$ | $53.65 \pm 45.74$ | $35.92 \pm 20.46$ | $40.14 \pm 10.67$ |
> | MRDR | $6.50 \pm 0.82$ | $7.53 \pm 4.77$ | $30.07 \pm 21.28$ | $15.94 \pm 8.59$ |
> | MIS | $65.50 \pm 37.68$ | $27.62 \pm 9.18$ | $18.14 \pm 5.58$ | $23.87 \pm 7.12$ |
> | Shaped\_PDIS | $354.72 \pm 746.17$ | $35.51 \pm 22.12$ | $18.75 \pm 2.26$ | $14.58 \pm 8.98$ |
> | Shaped\_DR | $3.21 \pm 0.58$ | $3.24 \pm 0.35$ | $1.85 \pm 0.27$ | $1.46 \pm 0.67$ |
> | Shaped\_MRDR | $1.53 \pm 1.74$ | $1.63 \pm 1.112$ | $3.06 \pm 2.63$ | $4.00 \pm 1.89$ |
>
> ### AntMaze MSE
>
> | Method | N=0.5k | N=1k | N=2k | N=5k |
> |--------|-------|--------|--------|--------|
> | PDIS | $134.68 \pm 210.64$ | $123.38 \pm 126.89$ | $75.30 \pm 64.85$ | $27.98 \pm 25.02$ |
> | DR | $49.79 \pm 45.44$ | $26.97 \pm 19.06$ | $7.81 \pm 6.01$ | $10.25 \pm 8.89$ |
> | MRDR | $9.38 \pm 5.34$ | $8.87 \pm 3.21$ | $11.08 \pm 12.73$ | $7.18 \pm 1.35$ |
> | MIS | $26.18 \pm 18.73$ | $29.52 \pm 22.47$ | $25.36 \pm 11.16$ | $19.24 \pm 8.25$ |
> | Shaped\_PDIS | $17.26 \pm 3.83$ | $10.17 \pm 8.37$ | $13.71 \pm 6.22$ | $13.61 \pm 2.10$ |
> | Shaped\_DR | $13.14 \pm 4.66$ | $13.45 \pm 2.85$ | $14.04 \pm 1.30$ | $14.98 \pm 0.88$ |
> | Shaped\_MRDR | $1.07 \pm 1.01$ | $1.33 \pm 0.81$ | $1.37 \pm 0.65$ | $1.14 \pm 0.27$ |
>
> Q2: The population-level guarantees for Shaped-DR and Shaped-MRDR are identical in structure to Shaped-PDIS and already appear in Appendix B (Theorems B.2–B.6) and Appendix C.1 (Equations 26–31). All three share the form $\hat{V}^{\pi\_e}\_{\text{Shaped}}(\lambda) = \frac{1}{N} \sum\_{n} [ Y^{(n)}\_{\text{base}} + \lambda^\top \tilde{C}^{\Phi,(n)} ]$ where $Y\_{\text{base}} \in \{Y\_{\text{PDIS}}, Y\_{\text{DR}}, Y\_{\text{MRDR}}\}$. Since $\tilde{C}^{\Phi}$ is zero-mean by construction (Lemma 4.2), all guarantees hold uniformly across variants: unbiasedness for any $\lambda$, quadratic variance decomposition, closed-form optimum $\lambda^\* = -\Sigma\_{CC}^{-1} \Sigma\_{CY}^{\text{base}}$, and monotone variance reduction $\text{Var}(Y\_{\text{base}} + \lambda^{\*\top} \tilde{C}^{\Phi}) \leq \text{Var}(Y\_{\text{base}})$. Only $Y\_{\text{base}}$ and consequently $\Sigma\_{CY}^{\text{base}}$ change across variants. We will consolidate these into a unified theorem in the camera-ready version.
> **Finite-sample behavior.** With estimated $\hat{\lambda}$, the excess variance $(\hat{\lambda} - \lambda^\*)^\top \Sigma\_{CC} (\hat{\lambda} - \lambda^\*) = O\_p(1/N)$ vanishes asymptotically and is further stabilized by ridge regularization. Even with poor $\Phi\_\beta$ estimates, the control variate remains zero-mean by construction, preserving unbiasedness unconditionally. We will add a detailed derivation in the camera-ready version.
>
> W3, Q3: We conduct additional ablations w.r.t. Architecture size, learning rate, and no. of training trajectories used to train the shaping function. All the results are for N=5k test samples over 10 seeds, while the remaining setup being identical. We observe, the results mostly stay consistent across various architectures, learning rates, and training trajectories higher than 1500.
>
> ### Architecture Ablations: PointMaze MSE for N=5k
>
> | Method | 1 layer 128 | 1 layer 256 | 2 layer 128-128 | 2 layer 256-256 |
> |--------|------------|------------|----------------|----------------|
> | Shaped\_PDIS | $15.20 \pm 6.83$ | $14.85 \pm 11.09$ | $14.58 \pm 8.98$ | $15.57 \pm 7.56$ |
> | Shaped\_DR | $1.45 \pm 0.57$ | $1.42 \pm 0.67$ | $1.46 \pm 0.67$ | $1.43 \pm 0.56$ |
> | Shaped\_MRDR | $4.26 \pm 1.45$ | $4.17 \pm 2.28$ | $4.10 \pm 1.71$ | $4.00 \pm 1.89$ |
>
> ### Learning Rate Ablations: PointMaze MSE for N=5k
>
> | Method | 1e-2 | 5e-3 | 1e-3 | 5e-4 | 1e-4 |
> |--------|------|------|------|------|------|
> | Shaped\_PDIS | $18.73 \pm 10.75$ | $14.58 \pm 8.98$ | $15.34 \pm 7.72$ | $15.30 \pm 9.16$ | $15.11 \pm 7.76$ |
> | Shaped\_DR | $2.57 \pm 0.74$ | $1.46 \pm 0.67$ | $1.52 \pm 0.72$ | $1.43 \pm 0.67$ | $1.56 \pm 0.65$ |
> | Shaped\_MRDR | $5.38 \pm 2.28$ | $4.00 \pm 1.89$ | $3.94 \pm 2.07$ | $3.92 \pm 1.92$ | $3.97 \pm 2.07$ |
>
> ### No. of Training Trajectories Ablations: PointMaze MSE for N=5k
>
> | Method | 1k | 1.5k | 2k | 5k |
> |--------|------|------|------|------|
> | Shaped\_PDIS | $39.27 \pm 11.74$ | $16.95 \pm 10.16$ | $14.20 \pm 9.06$ | $14.28 \pm 8.59$ |
> | Shaped\_DR | $22.06 \pm 0.79$ | $1.64 \pm 0.73$ | $1.40 \pm 0.70$ | $1.47 \pm 0.66$ |
> | Shaped\_MRDR | $15.84 \pm 2.21$ | $3.66 \pm 2.09$ | $3.98 \pm 2.07$ | $3.90 \pm 1.82$ |

---

> > ### Author Rebuttal · Reviewer_7ZQD · 2026-04-04
> >
> > I thank the authors for the rebuttal. The additional results for PointMaze and AntMaze and the sensitivity analysis  helpfully clarify the method's practical utility and stability. I encourage the authors to ensure that the detailed derivations for the finite-sample excess variance and the consolidated theorems are fully integrated into the final manuscript.

---

> > > ### Author Response · Authors · 2026-04-04
> > >
> > > We thank the reviewer for acknowledging our rebuttal and help us significantly strengthen our paper and making it more holistic. We will make the additions in the camera-ready version of the paper. If you are satisfied with the rebuttals and the paper's contributions, both from a theoretical and a practical standpoint, would you consider raising the score?
> > >
> > > ---
> > >
> > > Additionally, we thank the reviewer for carefully examining the references. We take reference accuracy extremely seriously, and we acknowledge and sincerely apologize for the error in the submitted version.
> > >
> > > The submitted version incorrectly cited:
> > >
> > > 1. Grzes, M. Reward shaping in episodic reinforcement learning. Artificial Intelligence, 236:1–30, 2016
> > > 2. Uehara, M., Shi, C., and Kallus, N. Review of off-policy evaluation in reinforcement learning. arXiv preprint arXiv:2204.05440, 2022.
> > >
> > > This was a misattribution error; while the paper title and the author list is correct, we had a corrupted citation metadata where the arxiv identifier belonged to a different paper. This was an honest mistake on your end, and we have proactively corrected all the citations in our paper. The corrected citations:
> > >
> > > 1. Grzes, M. Reward shaping in episodic reinforcement learning. In Proceedings of the 16th Conference on
> > > Autonomous Agents and MultiAgent Systems, AAMAS’17
> > > 2. Uehara, M., Shi, C., and Kallus, N. Review of off-policy evaluation in reinforcement learning. arXiv preprint **arXiv:2212.06355**.
> > >
> > > This has been corrected in the revised submission. We have also conducted a **full audit of references** in the paper.
> > >
> > > ---
> > >
> > > Full corrected reference list (short form):
> > >
> > > 1. Cassel, C. M., Sarndal, C. E., and Wretman, J. H. Some results on generalized difference estimation and generalized
> > > regression estimation for finite populations. Biometrika, 63(3):615–620, 1976.
> > >
> > > 2. Che, F., Chan, B., Ma, C., and Mahmood, A. R. AVGDICE: Stationary distribution correction by regression.
> > > Reinforcement Learning Journal, 6:2415–2426, 2025.
> > >
> > > 3. Choudhary, K., Gupta, D., and Thomas, P. S. ICU-Sepsis: A benchmark MDP built from real medical data. In RLC, 2024.
> > >
> > > 4. Devlin, S. and Kudenko, D. Dynamic potential-based reward shaping. In AAMAS, 2012.
> > >
> > > 5. Dudik, M., Langford, J., and Li, L. Doubly robust policy evaluation and learning. In ICML, 2011.
> > >
> > > 6. Farajtabar, M., Chow, Y., and Ghavamzadeh, M. More robust doubly robust off-policy evaluation. In ICML, pp. 1447–1456. PMLR, 2018.
> > >
> > > 7. **Grzes, M. Reward shaping in episodic reinforcement learning. In Proceedings of the 16th Conference on
> > > Autonomous Agents and MultiAgent Systems, AAMAS’17**
> > >
> > > 8. Hanna, J., Stone, P., and Niekum, S. Bootstrapping with models: Confidence intervals for off-policy evaluation. In AAAI, 2017.
> > >
> > > 9. Jiang, N. and Li, L. Doubly robust off-policy value evaluation for reinforcement learning. In ICML, 2016.
> > >
> > > 10. Le, H., Voloshin, C., and Yue, Y. Batch policy learning under constraints. In ICML, 2019.
> > >
> > > 11. Liu, Xiao-Yang et al. Finrl-meta: A universe of near-real market environments for data-driven deep reinforcement learning in quantitative finance, 2022.
> > >
> > > 12. Majumdar, R., Teversham, J., and Parbhoo, S. Concept-based off-policy evaluation. Reinforcement Learning
> > > Journal, 6:1951–1989, 2025.
> > >
> > > 13. Muller, H. and Kudenko, D. Improving the effectiveness of potential-based reward shaping in reinforcement learning, 2025.
> > >
> > > 14. Nachum, O., Chow, Y., Dai, B., and Li, L. Dualdice: Behavior-agnostic estimation of discounted stationary
> > > distribution corrections. In NeurIPS, 2019.
> > >
> > > 15. Ng, A. Y., Harada, D., and Russell, S. J. Policy invariance under reward transformations: Theory and application to reward shaping. In ICML, 1999.
> > >
> > > 16. Parbhoo, S., Gottesman, O., and Doshi-Velez, F. Shaping control variates for off-policy evaluation. In Offline RL Workshop at NeurIPS, pp. 93, 2020.
> > >
> > > 17. Precup, D., Sutton, R. S., and Singh, S. Eligibility traces for off-policy policy evaluation. In ICML, 2000.
> > >
> > > 18. Ribba, B. et al. A tumor growth inhibition model for low-grade glioma treated with chemotherapy or radiotherapy. Clinical Cancer Research, 18(18):5071–5080, 2012.
> > >
> > > 19. Shewchuk, J. R. et al. An introduction to the conjugate gradient method without the agonizing pain, 1994.
> > >
> > > 20. Thomas, P. S. and Brunskill, E. Data-efficient off-policy policy evaluation for reinforcement learning. In ICML, 2016.
> > >
> > > 21. Uehara, M., Huang, J., and Jiang, N. Minimax weight and q-function learning for off-policy evaluation. In ICML, 2020.
> > >
> > > 22. **Uehara, M., Shi, C., and Kallus, N. Review of off-policy evaluation in reinforcement learning. arXiv preprint arXiv:2212.06355, 2022**.
> > >
> > > 23. Zhang, S., Dai, B., Li, L., and Schuurmans, D. Gendice: Generalized offline estimation of stationary distribution
> > > corrections. In ICLR, 2020.
> > >
> > > ---
> > >
> > > Post the complete audit on references, would you consider increasing the score based on technical content + questions addressed through the rebuttal? We once again thank you for significantly improving and strengthing our paper.

---

### Decision · Program_Chairs · 2026-04-30

**Decision:**

Accept (regular)

**Comment:**

The reviewers find this a solid paper with several notable strengths, including:
1) Significance: The paper makes a contribution to off-policy evaluation, which is an important challenge in reinforcement learning
2) Correctness: The paper provides meaningful theoretical guarantees for the proposed algorithm
3) Originality: The ideas are novel and positioned well with respect to previous work.

However, the reviewers also identify several issues. None of them appears to be serious and can be addressed in the revision.
1) The paper comes off as a little sloppy, with an incorrect running title and errors in the citations noted by the reviewers. It is not clear whether this is a human error or the citations were hallucinated, but we urge the authors to review the paper carefully.
2) The empirical evaluation has some shortcomings, but these can be addressed in the camera-ready version. f

The authors also mention additional theoretical results in their response. Unless the additions are minor, they should go through peer review before publication.